# The shelterin component TRF2 mediates columnar stacking of human telomeric chromatin

Sook Yi Wong [1,3,5], Aghil Soman[1,5], Nikolay Korolev [1], Wahyu Surya [1], Qinming Chen[1,4], Wayne Shum[1], John van Noort[1,2] & Lars Nordenskiöld [1✉]

## Abstract

**Telomere repeat binding factor 2 (TRF2) is an essential component of the telomeres and also plays an important role in a number of other non-telomeric processes. Detailed knowledge of the binding and interaction of TRF2 with telomeric nucleosomes is limited. Here, we study the binding of TRF2 to in vitro-reconstituted kilo-basepair-long human telomeric chromatin fibres using electron microscopy, single-molecule force spectroscopy and analytical ultracentrifugation sedimentation velocity. Our electron microscopy results revealed that full-length and N-terminally truncated TRF2 promote the formation of a columnar structure of the fibres with an average width and compaction larger than that induced by the addition of $Mg^{2+}$, in agreement with the in vivo observations. Single-molecule force spectroscopy showed that TRF2 increases the mechanical and thermodynamic stability of the telomeric fibres when stretched with magnetic tweezers. This was in contrast to the result for fibres reconstituted on the 'Widom 601' high-affinity nucleosome positioning sequence, where minor effects on fibre stability were observed. Overall, TRF2 binding induces and stabilises columnar fibres, which may play an important role in telomere maintenance.**

**Keywords** Chromatin Structure; Electron Microscopy; Nucleosome; Shelterin Complex; Telomere
**Subject Categories** Chromatin, Transcription & Genomics; Structural Biology

## Introduction

The telomere is a nucleoprotein complex that serves as a protective cap against chromosomal end fusion, degradation, and amiss DNA repairs at the ends of linear eukaryotic chromosomes (Webb et al, 2013). The telomere is comprised of a repetitive sequence of concatenated 5–8 bp repeats with 3–4 consecutive guanines, forming a guanine-rich strand. In humans, the telomeric DNA is a 6-bp repeat of $(TTAGGG)_n$ of 5–15 kb in length (Samassekou et al, 2010). Telomere dysfunction is associated with tumour development and ageing pathologies (d'Adda di Fagagna et al, 2003; Hewitt et al, 2012; Pisano et al, 2008). Just as in other higher eukaryotes, the human telomeric repeats are organised with histone proteins as an array of nucleosomes that condenses into chromatin fibres (Fajkus et al, 1995; Lejnine et al, 1995; Makarov et al, 1993; Tommerup et al, 1994) which, together with the shelterin complex, form the capping structure (de Lange, 2005). The shelterin complex comprises the six protein subunits: TRF1, TRF2, POT1, RAP1, TIN2, and TPP1 (de Lange, 2018; Lim and Cech, 2021).

Telomeric chromatin has been perceived to be structurally and functionally different from bulk chromatin due to its repetitive sequence and interactions with the unique telomere-binding proteins (Cacchione et al, 1997; de Lange, 2005; Ichikawa et al, 2014; Pisano et al, 2008; Pisano et al, 2007; Rossetti et al, 1998). Nuclease digestion of telomeric chromatin revealed a much shorter nucleosome repeat length (NRL) of roughly $157 \pm 30$ bp compared to the longer NRLs of bulk chromatin (e.g. 197 bp in mammals) (Bedoyan et al, 1996; Makarov et al, 1993). Both the Cryo-EM and the crystal structure of nucleosome core particle (NCP) reconstituted on telomeric DNA, although displaying slightly different structural features, exhibit general architectures similar to NCP structures with positioning DNA sequences (Hu et al, 2023; Soman et al, 2020), but in solution, the telomeric NCP is more dynamic and less stable (Soman et al, 2020). We recently revealed a novel columnar arrangement of human telomeric chromatin fibres obtained by in vitro reconstitution of histone octamers (HOs) on telomeric (TTAGGG) DNA array templates folded in the presence of $Mg^{2+}$ (Soman et al, 2022b). We determined the cryo-EM structure of the compact columnar telomeric tetranucleosome and identified the crucial importance of the histone tails in mediating the closely stacked compact columnar fibre arrangement (Soman et al, 2022b).

Telomeric DNA acts as the platform for the recruitment of the telomere-specific DNA binding factors TRF1 and TRF2, as well as the other telomere proteins, which form the protective shelterin complex. Although there is a wealth of information on the functional role of specific proteins that bind at telomeres, little is known about telomeric chromatin and its interaction with the

[1]School of Biological Sciences, Nanyang Technological University, Singapore 637551, Singapore. [2]Huygens-Kamerlingh Ones Laboratory, Leiden University, Leiden 2333 AL, The Netherlands. [3]Present address: Department of Emerging Infectious Diseases, Duke-NUS, Medical School, Singapore 169857, Singapore. [4]Present address: M Diagnostics PTE. LTD, 30 Biopolis Street, Matrix, Singapore 138671, Singapore. [5]These authors contributed equally: Sook Yi Wong, Aghil Soman. ✉E-mail: larsnor@ntu.edu.sg

shelterin components at the molecular level (Galati et al, 2013; Lim and Cech, 2021; Mandemaker and Mattiroli, 2022). In addition, how this specialised chromatin may differ in structure and dynamics from 'canonical' chromatin has until recently been unknown (Mandemaker and Mattiroli, 2022; Soman et al, 2022a; Soman et al, 2022b). The discovery of the columnar form of telomeric chromatin, compacted in the presence of $Mg^{2+}$, has raised questions on the interaction of shelterin factors and their effects on telomeric chromatin structure.

Telomere repeat binding factor 2 (TRF2) comprises three functional domains: (1) the N-terminal basic domain, (2) the central TRFH domain, and (3) the C-terminal Myb/SANT domain (Broccoli et al, 1997). The basic N-terminus protects the T-loop structure and inhibits recombination endonuclease activity at the telomere (Necasová et al, 2017; Saint-Léger et al, 2014), while the TRFH domain induces TRF2 homodimerisation and acts as a recruitment site for various proteins to establish proper telomere function. The Myb/SANT domain facilitates the binding of homodimer TRF2 to the double-stranded telomeric DNA (Court et al, 2005). TRF2 is a vital component of the shelterin complex for telomere protection by promoting the formation of the T-loop structure (Doksani et al, 2013; Griffith et al, 1999), repressing canonical non-homologous end-joining (Bae and Baumann, 2007; Doksani et al, 2013; Ribes-Zamora et al, 2013), and ataxia telangiectasia mutated signalling pathway (Denchi and de Lange, 2007).

TRF2 has been shown to down-regulate nucleosome occupancy and increase nucleosome spacing, as demonstrated by the over-expression of TRF2 in mice and humans (Bae and Baumann, 2007; Ribes-Zamora et al, 2013), which can be recapitulated in vitro (Galati et al, 2012). On the other hand, an AFM study showed that TRF2 binding to reconstituted nucleosome arrays resulted in the compaction of the nucleosome fibre (Baker et al, 2011). Previous in vitro studies of TRF2 with nucleosome arrays were performed on DNA with mixed sequences where some DNA templates were non-telomeric (Baker et al, 2011; Galati et al, 2012; Galati et al, 2015). These contradictory observations (negative and positive effects on nucleosome occupancy and chromatin compaction) and the recent discovery of the columnar structure of telomeric chromatin (Soman et al, 2022b) motivated us to further investigate the structural and biophysical properties of telomeric chromatin in the presence of TRF2.

Here, we study the binding of the N-terminal truncated TRF2 (TRF2$^{\Delta N}$), complemented with data for the full-length TRF2 protein to well-defined telomeric nucleosome arrays to establish its effect on the telomeric chromatin structural and dynamic properties. Importantly, we observe that TRF2 alone is sufficient to induce a homogeneous fibre arrangement by compacting the telomeric array into the tight columnar architecture similar to the published cryo-EM structure of the telomeric tetranucleosome (Soman et al, 2022b), resulting in more compact fibres with a significantly larger average fibre diameter compared to compaction in the presence of $Mg^{2+}$. Furthermore, the characterisation of the compacted telomeric nucleosome arrays using single molecule force spectroscopy demonstrated pronounced changes in the mechanical and thermodynamic properties of the fibres upon TRF2 binding that stabilise the telomeric chromatin, which was not displayed for nucleosome arrays reconstituted with the '601' high affinity nucleosome positioning sequence.

# Results

## Production of telomeric DNA and telomeric nucleosome arrays

We used our previously developed general and flexible method to prepare large quantities of telomeric DNA with 157 bp repeat length comprising telomeric repeats (Soman et al, 2022b). In brief, repeats of 157 bp telomeric DNA consisting entirely of human telomeric TTAGGG repeats flanked by AvaI restriction sites were multimerised to produce a DNA array containing 10×157 bp telomeric DNA (Telo-10 DNA) (Appendix Fig. S1A). For the magnetic tweezers experiments, a 20-mer hybrid construct with 18 repeats of 157 bp telomeric DNA flanked by 157 bp of '601' nucleosome positioning DNA sequence (Thåström et al, 2004) was also produced (#Telo-18# DNA; '#' indicates 157 bp of the '601' DNA) (Soman et al, 2022b). We studied the TRF2 interaction with '601' (601-10 and 601-20) and telomeric nucleosome arrays, reconstituted using recombinant human histones (Appendix Fig. S1A–H). This enabled us to establish the sequence preferences of the TRF2 protein by comparison of the data obtained for the telomeric constructs compared to '601' arrays (Soman et al, 2022b). A description of sample preparation and the methods used in the studies are given in the 'Methods' section.

## TRF2$^{\Delta N}$ binding to telomeric nucleosome array does not cause histone eviction

In order to investigate the structural and functional importance of TRF2 protein has on telomeric chromatin, a stable TRF2-chromatin complex is preferred in downstream studies using analytical ultracentrifugation (AUC), electron microscopy and single-molecule force spectroscopy. Previous studies have shown that the removal of the N-terminal basic domain formed a nucleosome complex with higher binding affinity than full-length TRF2 (Galati et al, 2015). Moreover, a recent LLPS study showed that truncation of the N-terminal domain of TRF2 does not significantly change its propensity to induce liquid-liquid phase separation (LLPS) of telomeric DNA, and its binding to the telomeric DNA remains sufficiently favourable (Jack et al, 2022). Our study mainly employed the N-terminal truncated TRF2 (aa 42-524), which herein is termed as TRF2$^{\Delta N}$, complemented with full-length TRF2 (denoted 'TRF2'), which showed the same structural features as the former in EM imaging of the structural effects of the protein binding to telomeric chromatin (see below).

Previous studies of TRF2 binding to DNA and nucleosome arrays were performed under near-physiological conditions containing high concentrations of monovalent and divalent cations (Baker et al, 2011; Galati et al, 2012; Galati et al, 2015). As a high concentration of cations can fold and compact individual arrays and bring about the inter-array association (Allahverdi et al, 2015; Allahverdi et al, 2011; Hansen et al, 1989), we first assessed and compared the effect of cations on TRF2$^{\Delta N}$ binding to the nucleosome array. Three conditions were compared: (1) low monovalent salt concentration (5 mM NaCl), (2) moderate to high monovalent salt concentration (62.5–150 mM NaCl) and (3) near-physiological salt concentrations using Telo-10 and 601-10 or 601-20 array titrated with increasing concentration of the TRF2$^{\Delta N}$ dimer (Fig. EV1). The EMSA results showed an insignificant difference in

terms of band migration under the tested salt conditions, which ruled out the unspecific interaction of the TRF2$^{\Delta N}$ protein to the nucleosome arrays at low salt conditions. As such, we used 5 mM NaCl (low monovalent salt condition) for all EMSA, AUC, and EM experiments (see below) to eliminate any possible salt-induced array compaction and examine the effect on the nucleosome array upon TRF2$^{\Delta N}$ binding.

The addition of the TRF2$^{\Delta N}$ dimer leads to a slower band migration of the Telo-10 and 601-10 arrays, which indicates the binding of TRF2$^{\Delta N}$. The band shift of the array was observed with as little as 0.13 µM TRF2$^{\Delta N}$ dimer (corresponding to 0.4 TRF2$^{\Delta N}$ dimer/157 bp DNA) added to both Telo-10 and 601-10 arrays. Notably, the 601-10 array showed more distinct and clear bands than the Telo-10 array, likely because the telomeric nucleosome arrays, on their own, are more heterogeneous compared to the 601-Widom arrays that are designed with a strong nucleosome positioning sequence (compare lane 3 of Fig. EV1A,C,E with lane 3 of Fig. EV1B,D,F) (Soman et al, 2022b). Based on these results, we suggested similar binding of TRF2$^{\Delta N}$ to the Telo-10 and 601-10 or 601-20 arrays. We also observed a similar binding pattern, with more distinct bands, when TRF2$^{\Delta N}$ was bound to Telo-10 and 601-10 DNA (Fig. EV2A,B). Closer inspection showed that TRF2$^{\Delta N}$ has a slight preference for binding to Telo-10 DNA (marked with red diamonds), suggesting a higher affinity of TRF2$^{\Delta N}$ for telomeric sequence. Hence, we questioned whether removing the N-terminus of TRF2 reduced or eliminated TRF2 specificity for telomeric sequence by performing a competition binding assay (Fig. EV2C). With an equal amount of telomeric and 601-Widom DNAs (Telo-10 and 601-5 DNA), we observed that TRF2$^{\Delta N}$ remained highly specific for telomeric sequence, whereby the binding of TRF2$^{\Delta N}$ to 601-5 DNA was only observed after Telo-10 DNA was saturated, in agreement with previous literature, using full-length TRF2 (Baker et al, 2011). Unspecific binding of TRF2 to non-telomeric regions may not be surprising given that TRF2 has been shown to bind to other sequences (e.g. with G-quadruplex motifs) to functionally regulate the expression of several genes through epigenetic modification on the promoter (Imran et al, 2021; Mukherjee et al, 2019). TRF2 also plays a role in neuronal gene silencing, stem cell fate and cancer cell survival through its interaction with repressor element 1-silencing transcription factor (REST), though direct interaction with DNA has not been implicated, we speculate that the ability of TRF2 to bind non-telomeric sequences plays a role in this (Kwon et al, 2013; Zhang et al, 2008). Moreover, in in vivo conditions, TRF2 also interacts with other shelterin components, such as Rap1, to increase its specificity for telomeric sequences while reducing its binding affinity for the other sequences (Janoušková et al, 2015).

Since TRF2 recognises and binds to the telomeric DNA through its Myb/SANT domain, we asked if TRF2 binding would cause the eviction of histone proteins from the nucleosome array to gain access to the telomeric DNA. To answer this question, we reversibly precipitated the TRF2$^{\Delta N}$-Telo-10 nucleosome array complex with 5 mM MgCl$_2$ (Appendix Fig. S2). If the addition of TRF2$^{\Delta N}$ resulted in the eviction of histone protein, histone proteins would not be precipitated and would remain in the supernatant fraction. Instead, we observed no histone proteins in the supernatant fraction, even at high concentrations of TRF2$^{\Delta N}$ dimer. This suggests that TRF2$^{\Delta N}$ binding to the telomeric array does not evict histone proteins.

## TRF2$^{\Delta N}$ binds and compacts telomeric and '601' nucleosome arrays differently

Next, we investigate the effect of TRF2$^{\Delta N}$ binding on telomeric chromatin fibres using the AUC-SV method. The AUC-SV experiment provides hydrodynamic information, such as the molecular mass and shape of a macromolecule based on the rate of the macromolecule sedimentation. In nucleosome array studies, the peak shape reflects the sample homogeneity and conformation distribution of nucleosome fibres, while the peak shift reflects the level of compaction of the fibres (Allahverdi et al, 2015; Dorigo et al, 2003; Fletcher and Hansen, 1995; Korolev et al, 2010) or a change in mass due to a change in nucleosome occupancy. We used this method to investigate the influence of TRF2$^{\Delta N}$ on the nucleosome array structure. This method has been widely used to study nucleosome array compaction induced by the addition of Mg$^{2+}$ to extended chromatin fibres prepared from Widom '601' positioning sequences. An increase of the Mg$^{2+}$ concentration results in a gradual folding of individual arrays until the fibres reach the maximally folded state, while further increase results in array self-association and aggregation (Allahverdi et al, 2011; Hansen et al, 1989).

At low salt and in the absence of Mg$^{2+}$, the extended Telo-10 array has a sedimentation coefficient $s_{20,w} = 27.9 \pm 0.4$ S, displaying an $s_{20,w}$ distribution with a full width at half maximum FWHM = 6.7 S (Fig. 1A top panel, black line). The addition of 0.2 µM TRF2$^{\Delta N}$ dimer (corresponding to 0.6 TRF2$^{\Delta N}$ dimer/157 bp DNA) to Telo-10 arrays resulted in an $s_{20,w}$ increase to $35.4 \pm 0.7$ S that displayed a narrower $s_{20,w}$ distribution, with the FWHM reduced from 6.7 S to 4.4 S (Fig. 1A top panel, green line). An increase in $s_{20,w}$ and a decrease in FWHM imply that the Telo-10 arrays in the complex with TRF2$^{\Delta N}$ are more compact and homogeneous than the extended and flexible Telo-10 array alone.

For comparison, we conducted the AUC-SV experiments on the Telo-10 array in the presence of Mg$^{2+}$ (Fig. 1A lower panel). The addition of 0.2 mM Mg$^{2+}$ to the Telo-10 arrays resulted in a shift in $s_{20,w}$ value from $27.3 \pm 0.6$ S (lower panel, black line) to $32.0 \pm 1.4$ S (lower panel, orange line), which indicates array compaction upon the addition of the divalent cation. The $s_{20,w}$ distribution was broadened in the presence of Mg$^{2+}$, and the FWHM value increased from 6.6 S to 10.0 S in the Telo-10 array. A broadening $s_{20,w}$ distribution implies an increase in heterogeneity in the conformation of compacted fibres.

A shift in the $s_{20,w}$ values can be attributed to a change in molecular weight or structural conformation. It was shown (Soman et al, 2022b) that the increase in the $s_{20,w}$ value upon adding Mg$^{2+}$ to the telomeric arrays resulted from compaction to the columnar form of telomeric chromatin fibres. In the case of adding TRF2$^{\Delta N}$ to the nucleosome array, the increase in the $s_{20,w}$ could result from a molecular weight change (due to the additional TRF2$^{\Delta N}$ proteins) and/or a change in the structural conformation. Nonetheless, the $s_{20,w}$ distribution was the narrowest in the TRF2$^{\Delta N}$-Telo array complex, compared to the extended Telo-10 array and Telo-10 arrays compacted with Mg$^{2+}$, suggesting that TRF2$^{\Delta N}$-Telo array complex had more homogeneous structural distribution.

As a control, we conducted the AUC-SV experiments on the 601-10 arrays in the presence of TRF2$^{\Delta N}$ as well as Mg$^{2+}$ (Fig. 1B). The extended 601-10 array has an $s_{20,w}$ value of $31.4 \pm 0.5$ S with a narrow $s_{20,w}$ distribution, FWHM = 3.7 S (Fig. 1B top panel, black

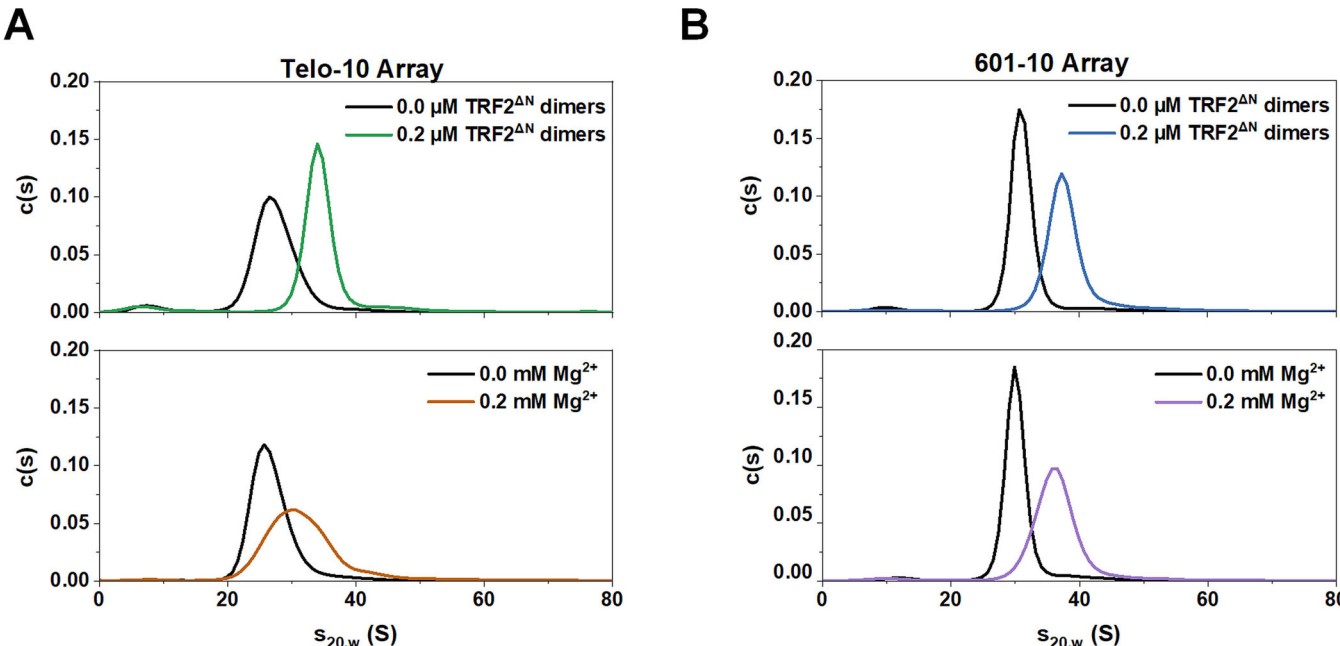

**Figure 1.  TRF2^ΔN binding to telomeric nucleosome array induces homogeneous fibre conformation.**

(**A**) Sedimentation coefficient (c(s)) distribution curves, obtained from AUC-SV data, for Telo-10 array in the presence of TRF2$^{\Delta N}$ (top) and Mg$^{2+}$ (bottom). Top panel: The s-value of the Telo-10 array in the absence of TRF2$^{\Delta N}$ dimers was 27.9 ± 0.4 S (black), which shifted to 35.4 ± 0.7 S in the presence of 0.2 µM TRF2$^{\Delta N}$ dimers (corresponding to 0.6 TRF2$^{\Delta N}$ dimer/157 bp of DNA) (green). Bottom panel: The s-value of the Telo-10 array at 0.0 mM Mg$^{2+}$ was 27.3 ± 0.6 S (black), which shifted to 32.0 ± 1.4 S in the presence of 0.2 mM Mg$^{2+}$ (orange). The data shown are the average of the three technically replicated experiments. Data are means ± s.d. (**B**) c(s) distribution curves, obtained from AUC-SV data, for 601-10 array in the presence of TRF2$^{\Delta N}$ (top) and Mg$^{2+}$ (bottom). Top panel: The s-value of the 601-10 array in the absence of TRF2$^{\Delta N}$ dimer was 31.4 ± 0.5 S (black), which shifted to 38.9 ± 1.0 S in the presence of 0.2 µM TRF2$^{\Delta N}$ dimers (corresponding to 0.6 TRF2$^{\Delta N}$ dimer/157 bp of DNA) (blue). Bottom panel: The s-value of 601-10 array at 0.0 mM Mg$^{2+}$ was 30.7 ± 0.3 S (black), which shifted to 36.9 ± 0.9 S in the presence of 0.2 mM Mg$^{2+}$ (purple). The data shown are the average of the three technically replicated experiments. Data are means ± s.d. Source data are available online for this figure.

line), which is a reflection of the homogeneity of fibres prepared from the '601' positioning sequence (Allahverdi et al, 2011; Hansen et al, 1989). The addition of 0.2 µM TRF2$^{\Delta N}$ dimer (corresponding to 0.6 TRF2$^{\Delta N}$ dimer/157 bp DNA) to 601-10 arrays resulted in an $s_{20,w}$ value shift to 38.9 ± 1.0 S. The addition of TRF2$^{\Delta N}$ protein to the 601-10 array also resulted in a broadening of the $s_{20,w}$ distribution, with the FWHM value increasing from 3.7 S to 5.0 S (Fig. 1B top panel, blue line). As shown in Fig. 1B, the addition of 0.2 mM Mg$^{2+}$ to the 601-10 arrays resulted in a shift in $s_{20,w}$ values from 30.7 ± 0.3 S (Fig. 1B bottom panel, black line) to 36.9 ± 0.9 S (bottom panel, purple line), which indicates array compaction upon the addition of the divalent cation. The $s_{20,w}$ distribution was broadened in the presence of Mg$^{2+}$, with the FWHM value increased from 3.6 S to 6.2 S in the 601-10 array. This result is similar to the previous observation for '601' arrays (Allahverdi et al, 2011).

A shift to a higher $s_{20,w}$ values after the addition of Mg$^{2+}$ and TRF2$^{\Delta N}$ to 601-10 array reflects a change in molecular weight and/ or change in structural conformation. In both cases, $s_{20,w}$ distributions were broadened (increase in FWHM values), which signified that the 601-10 array compacted by Mg$^{2+}$ and TRF2$^{\Delta N}$-601 array complex was more heterogeneous than the extended 601-10 array. Comparing the four sets of data presented in Fig. 1A,B, it was noteworthy that only the TRF2$^{\Delta N}$-Telo array complex shows a narrowing of the $s_{20,w}$ distributions (decrease in FWHM value),

indicating that TRF2$^{\Delta N}$ induced homogeneous conformational changes on the Telo-10 fibre population that was not observed in other conditions. We hypothesised that this homogeneity effect when TRF2$^{\Delta N}$ binds to the Telo-10 array could be a result of one (or both in combination) of the two factors:

1. TRF2$^{\Delta N}$ arranges the histone octamer along the Telo-10 array to achieve a more homogeneous distribution of fibres (if the shift in $s_{20,w}$ value is mainly attributed to the change in molecular weight and the narrowing of $s_{20,w}$ distribution is due to the binding of TRF2$^{\Delta N}$ that assist in the repositioning of histone octamer in an orderly manner).
2. TRF2$^{\Delta N}$ promotes homogeneous compaction of the Telo-10 array (if the shift in $s_{20,w}$ value is mainly attributed to the change of fibre conformation, and the narrowing of $s_{20,w}$ distribution is due to a more homogeneous fibre population).

Given that TRF2$^{\Delta N}$ binds to telomeric and 601-Widom DNA (Fig. EV2), it raised the question if the shift in s-value observed with Telo-10 and 601-10 array was an effect from the interaction of TRF2$^{\Delta N}$ with DNA. As such, we also conducted AUC-SV experiments on the Telo-10 and 601-10 DNAs in the presence and absence of TRF2$^{\Delta N}$ (Fig. EV2D). For both Telo-10 and 601-10 DNA, we observed a slight increase in s-value from 9.49 ± 0.02 S to 10.4 ± 0.2 S and 9.7 ± 0.02 S to 10.6 ± 0.03 S, respectively, in the presence of TRF2$^{\Delta N}$. More

importantly, the addition of TRF2$^{\Delta N}$ protein to the DNAs resulted in the broadening of the $s_{20,w}$ distribution, which implied heterogeneous populations of TRF2$^{\Delta N}$-DNA complex. The results suggest that the interaction of TRF2$^{\Delta N}$ alone with telomeric DNA is insufficient to promote the same homogeneous conformational changes observed with Telo-10 fibre population by TRF2$^{\Delta N}$, and the phenomenon of having narrower $s_{20,w}$ distribution occurs only in the presence of telomeric nucleosomes.

Taken together, TRF2$^{\Delta N}$ binding appears to have a different influence on the Telo-10 and 601-10 arrays since only the TRF2$^{\Delta N}$-Telo arrays showed the formation of a more homogeneous population (narrower $s_{20,w}$ distribution). Our AUC data suggested that the TRF2$^{\Delta N}$ binding promotes a more uniform conformational change unique to telomeric arrays and not observed in '601' arrays.

## TRF2$^{\Delta N}$ promotes homogeneous columnar arrangement in telomeric fibres

To validate the AUC-SV data and determine the reason for the $s_{20,w}$ shift and narrowing of $s_{20,w}$ distribution, we visualised the complexes of TRF2$^{\Delta N}$ with telomeric arrays using negative-stain EM to investigate its structural effects in comparison with '601' arrays (Fig. 2A–E) and analysed the structural features of the telomeric and '601' arrays (Fig. 3A–D).

TRF2$^{\Delta N}$ was incubated with the respective arrays in potassium cacodylate buffer (pH 6.0), a buffer condition we had previously optimised for visualising the native nucleosome array without using a fixative agent (Soman et al, 2022b). This increases the positive charge of histone proteins (+18e) and TRF2$^{\Delta N}$ (+6.3e). The higher positive charge on the histone proteins and TRF2$^{\Delta N}$ allow a strong interaction with the DNA molecules, thereby decreasing the DNA dynamics and strengthening the overall structural of the TRF2$^{\Delta N}$-array complex. As such, lower concentrations of Mg$^{2+}$ and TRF2$^{\Delta N}$ are required to condense the nucleosome fibre (Ekundayo et al, 2017; Soman et al, 2022b).

In low-salt buffer (20 mM potassium cacodylate and 1 mM DTT), both Telo-10 and 601-10 array adopted an extended beads-on-the-string conformation in the absence of TRF2$^{\Delta N}$ and Mg$^{2+}$ (Fig. EV3A,B), and no compaction of the nucleosome array (in terms of a ladder, globular or columnar structures) were observed. The addition of 0.046 µM TRF2$^{\Delta N}$ dimer (corresponding to 0.4 TRF2$^{\Delta N}$ dimer/157 bp DNA) to Telo-10 arrays led to fibre compaction dominated by a well-defined columnar architecture with neighbouring nucleosomes stacked on top of each other (Blow-ups 1 and 2 and white triangles in Figs. 2A and 3A). In comparison, the addition of 0.045 mM Mg$^{2+}$ to the Telo-10 arrays resulted in fibre compaction of various conformations that, in addition to columnar fibres (42.5%), also contained irregular globules (52.5%) (Figs. 2B and 3A). Throughout the micrograph in Fig. 2A, the classical wedge-like feature of a side-view nucleosome (marked with purple triangles) was clearly observed along the columnar fibres, which confirms that the columnar structure was the compacted nucleosome arrays rather than compacted naked telomeric DNA with TRF2$^{\Delta N}$. Moreover, the same feature was also observed in the Telo-10 array compacted with Mg$^{2+}$ in the absence of TRF2$^{\Delta N}$ (Fig. 2B), further proving the presence of nucleosomes along the columnar fibre.

The data from Mg$^{2+}$-compacted arrays also agree with our earlier results, which showed that telomeric fibres formed folded

structures dominated by the columnar arrangement in the presence of Mg$^{2+}$ (Soman et al, 2022b). Remarkably, our present results show that TRF2$^{\Delta N}$ alone was sufficient to induce homogeneous columnar compaction of the Telo-10 arrays with a larger fraction (82.5%) of compact columns relative to what was observed in the presence of Mg$^{2+}$ (Fig. 3A). This agrees with our AUC data (Fig. 1A,B), showing that the Telo-10 array adopts a more homogeneous conformation in the presence of TRF2$^{\Delta N}$ protein but not in the presence of Mg$^{2+}$.

Furthermore, to ascertain and quantify the level of compaction in the telomeric fibres, the inter-nucleosomal distance between the midpoints of two neighbouring nucleosomes within a fibre was measured (Fig. 3B). Extreme outliers with an inter-nucleosomal distance longer than 15 nm were omitted to avoid taking measurements of nucleosomes in an 'open-state' (Soman et al, 2022b). The result showed a statistically significant difference in inter-nucleosomal distance between TRF2$^{\Delta N}$ and Mg$^{2+}$ compacted fibres, demonstrating a shorter mean distance in the presence of TRF2$^{\Delta N}$ (7.4 ± 1.7 nm compared to 7.7 ± 1.5 nm, $p = 0.025$). In other words, the presence of TRF2$^{\Delta N}$ leads to more compact fibres characterised by closer inter-nucleosomal stacking. The results suggest that for both cases (fibres folded in the presence of Mg$^{2+}$ or TRF2$^{\Delta N}$), the stacking between nucleosomes corresponds to the closest and most stable arrangement that maximises the tail-mediated interactions that promote the columnar architecture (Soman et al, 2022b).

In our previous work, we determined the cryo-EM structure of the columnar form of the telomeric tetranucleosome compacted in the presence of Mg$^{2+}$. Here, using cryo-EM imaging, we also demonstrated that not only Telo-10 fibres but also the tetranucleosome adopts the columnar architecture in the presence of TRF2$^{\Delta N}$ (Fig. 2C). The higher resolution in the cryo-EM images (compared to negative stain) also further demonstrates that the TRF2$^{\Delta N}$-induced structural change of the telomeric chromatin fibres is the result of the formation of the columnar structure with nucleosomes stacked on top of each other. The nucleosomes in the compacted telomeric tetranucleosomes are clearly identified by the wedge-like form and the wrapped DNA with the DNA gyres that are clearly identified in the blow-up images (Fig. 2C).

On the other hand, 601-10 arrays were folded in a zig-zag ladder-like arrangement in the presence of TRF2$^{\Delta N}$ or Mg$^{2+}$ (Fig. 2D,E), similar to the previously published 157-601 tetranucleosome structure (Ekundayo et al, 2017). The proportion of fibres in the zig-zag conformation is similar in the presence of TRF2$^{\Delta N}$ (69%), or Mg$^{2+}$ (64.5%), and no columnar fibres are observed (Fig. 3C). Despite this, we noticed a distinct difference in compactness among the 601-10 arrays with the TRF2$^{\Delta N}$ protein, such that some arrays were tightly compacted (marked with red arrows in Fig. 2D), while other arrays appeared as loosely folded ladder fibres (marked with orange arrows in Fig. 2D). To ascertain and quantify the level of compaction in 601 fibres, the inter-nucleosomal distance between the midpoints of two neighbouring nucleosomes within a fibre was measured (Fig. EV3C). Indeed, the TRF2$^{\Delta N}$-folded 601-10 arrays had a broader inter-nucleosome distance distribution (15.1 ± 3.6 nm) compared to the Mg$^{2+}$-compacted 601-10 array (13.7 ± 3.1 nm; $p = 6.53 \times 10^{-9}$). This shows that the TRF2$^{\Delta N}$-folded 601-10 arrays were less compacted. In the presence of TRF2$^{\Delta N}$, Telo-10 arrays were folded with a columnar feature resembling a continuous one-start spiral model, while the

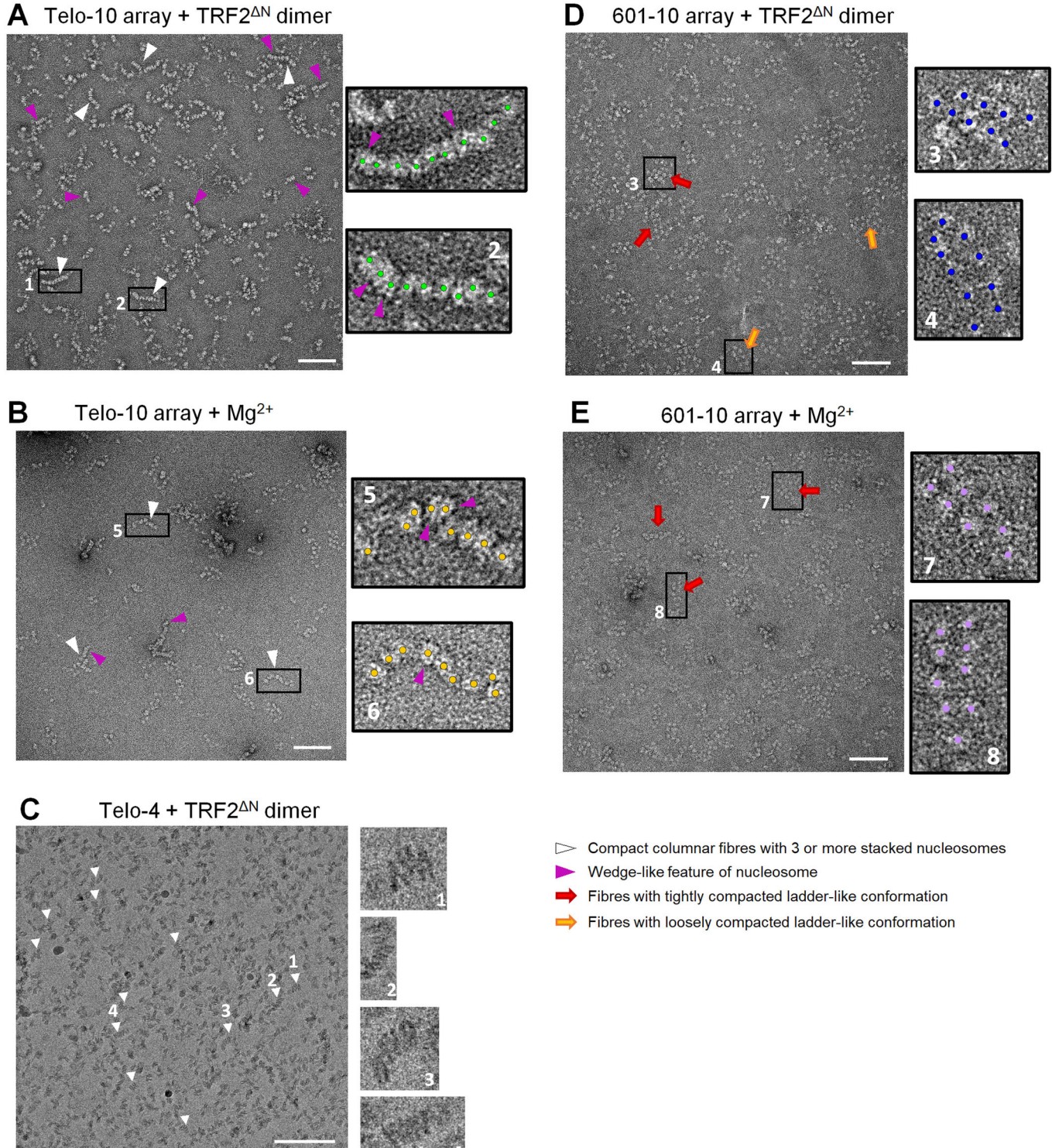

**A** Telo-10 array + TRF2<sup>ΔN</sup> dimer

**B** Telo-10 array + Mg²⁺

**C** Telo-4 + TRF2<sup>ΔN</sup> dimer

**D** 601-10 array + TRF2<sup>ΔN</sup> dimer

**E** 601-10 array + Mg²⁺

▷ Compact columnar fibres with 3 or more stacked nucleosomes
► Wedge-like feature of nucleosome
→ Fibres with tightly compacted ladder-like conformation
→ Fibres with loosely compacted ladder-like conformation

601-10 array was folded with 2 distinct columns of nucleosomes resembling a two-start zig-zag model.

Our previous work (Soman et al, 2022b) showed that the telomeric nucleosomes were organised with aligned minor and major grooves that formed a continuous 'supergroove' along the columnar fibre. This supergroove was hypothesised to be an ideal site for the interaction with telomeric proteins and chromatin remodellers. Although the resolution in negative stain EM images and the likely heterogeneity of the binding do not enable the TRF2<sup>ΔN</sup> protein to be identified, its potential binding to the supergrooves on the surface of the columnar form of telomeric chromatin is expected to increase the fibre diameter. As such, we

**Figure 2.  TRF2$^{\Delta N}$-Telomeric array complex adopted a homogeneous columnar arrangement.**

(A, D) Representative negative-stained EM micrographs of (A) Telo-10 array and (D) 601-10 array in the presence of 0.046 μM TRF2$^{\Delta N}$ dimers (corresponding to 0.4 TRF2$^{\Delta N}$ dimer/157 bp of DNA). Blow-ups 1 and 2 show columnar Telo-10 fibres in the presence of TRF2$^{\Delta N}$. Blow-ups 3 and 4 show ladder-like 601-10 fibres in the presence of TRF2$^{\Delta N}$. Scale bar: 100 nm. The data shown are representatives of three technically replicated experiments. (B, E) Representative negative-stained EM micrographs of (B) Telo-10 array and (E) 601-10 array in 0.045 mM Mg$^{2+}$. Blow-ups 5 and 6 show columnar Telo-10 fibres in the presence of Mg$^{2+}$. Blow-ups 7 and 8 show ladder-like 601-10 fibres in the presence of Mg$^{2+}$. Scale bar: 100 nm. The data shown are representatives of three technically replicated experiments. (C) Representative cryo-EM micrographs of Telo-4 in the presence of 0.5 TRF2$^{\Delta N}$ dimer/157 bp of DNA. Blow-ups 1–4 show columnar Telo-4 fibres in the presence of TRF2. Scale bar: 100 nm. The data shown are representatives of three technically replicated experiments. Source data are available online for this figure.

measured the mean fibre diameter of the Telo-10 array in the presence of TRF2$^{\Delta N}$ and Mg$^{2+}$ from a large number of compact fibres in the images (Fig. 3D). In the presence of TRF2$^{\Delta N}$, the fibre diameter has a mean value of 15.3 nm, approximately 1.6 nm wider than Mg$^{2+}$-compacted Telo-10 fibres, suggesting that TRF2$^{\Delta N}$ binds to the surface of the telomeric chromatin fibre resulting with a wider fibre diameter. These values were consistent with a recent in-vivo study (Hübner et al, 2022) of native chromatin at telomeres. That work showed that the fibre width, when measured by tagging with TRF2 (15.4 nm), was approximately 1 nm larger than the fibre width measured by tagging the H2B histone (14.4 nm) and suggested that TRF2 binds on the surface of the telomeric fibre.

Collectively, the result from negative-stain EM emphasised the conformational difference in the fibre compaction of Telo-10 and 601-10 arrays in the presence of TRF2$^{\Delta N}$ and Mg$^{2+}$. Remarkably, TRF2$^{\Delta N}$ alone is sufficient to generate a homogeneous compaction of the Telo-10 array with a columnar arrangement by binding on the surface of the telomeric chromatin, possibly along the continuous 'supergrooves'. The EM data also agreed with the AUC-SV results, showing similar $s_{20,w}$ distribution in the presence of both TRF2$^{\Delta N}$ and Mg$^{2+}$ that represented the conformational distribution of the samples.

The EM results described above demonstrated that TRF2$^{\Delta N}$ binding to telomeric chromatin fibres induces the columnar form, which was obtained with the N-terminal truncated TRF2 construct (TRF2$^{\Delta N}$); we wondered whether this result is a general feature that is also characteristic of the full-length TRF2. To this end, we prepared the full-length TRF2 (TRF2) (Appendix Fig. S1F) and performed negative stain EM imaging of Telo-10 fibres as well as cryo-EM imaging of Telo-4 arrays (Fig. EV3D–F) in the presence of TRF2. We observed that the TRF2 clearly induced the columnar conformation of telomeric chromatin with the compaction observed at a slightly lower ratio of TRF2 compared to TRF2$^{\Delta N}$ (Figs. 2A and EV3D,E). The results demonstrated unequivocally that the full-length TRF2 has the same effect and also leads to the formation of the columnar form of compact telomeric chromatin, which suggests that the compaction of telomeric chromatin into the columnar form is physiologically relevant for telomere compaction and maintenance.

## Single-molecule force spectroscopy reveals the influence of TRF2$^{\Delta N}$ binding on the mechanical properties of telomeric chromatin fibres at the single-molecule level

Having established that TRF2$^{\Delta N}$ induces the columnar architecture of telomeric chromatin and results in a larger fibre diameter, which we tentatively propose is caused by the binding of TRF2$^{\Delta N}$ to DNA supergrooves on the array, we asked how the TRF2$^{\Delta N}$ binding affects the mechanical and thermodynamic properties of the fibres.

To this end, we used single molecule force spectroscopy measurements using multiplexed magnetic tweezers (MMT). This method enables the analysis of folded single chromatin fibres without staining, fixation or surface deposition and to characterise the force-extension curve from sub-pN to tens of pN force, following the nucleosome unstacking to complete DNA unwrapping (Brouwer, 2020; Kaczmarczyk et al, 2018; Korolev et al, 2022; Kruithof et al, 2009; Meng et al, 2015; Soman et al, 2022b). Subsequent data analysis yields mechanical (stiffness) and thermodynamic (free energies of nucleosome unstacking and DNA unwrapping) parameters specific to the chromatin fibre under investigation. Here, we compare the stretching response of the compact chromatin fibres reconstituted on the telomeric and '601' DNA sequences under physiological salt conditions (100 mM KCl, 2 mM MgCl$_2$) in the presence and absence of the TRF2$^{\Delta N}$ dimer in the flow cell. We used nucleosome arrays reconstituted on the two different DNA templates: (1) #Telo-18# DNA comprising a Telo-18 (18 × 157) telomeric sequence flanked by one 157 bp '601' DNA on either side; (2) 601-20 DNA consisting of 20 157 bp NRL DNA of the '601' Widom sequence. The MMT study was carried out as described in 'Methods'.

The force-extension curves (Fig. 4A) of #Telo-18# and 601-20 arrays display similar features to previously studied nucleosome arrays (Brouwer et al, 2021; Kaczmarczyk, 2019; Korolev et al, 2022; Kruithof et al, 2009; Meng et al, 2015) with parameters in agreement with results for the identical arrays published earlier (Soman et al, 2022b). Increasing the stretching force leads to a series of transitions reflecting the unfolding of the compact nucleosome array and gradual DNA unpeeling from the histone core. These stages are schematically depicted in Fig. 4B and explained in the legend of this figure. Fitting the experimental points to the statistical mechanics model (Kaczmarczyk et al, 2018; Meng et al, 2015) provides information about the arrays' mechanical properties: The fitting gives the number of nucleosomes in the array ($N_{total}$) as well as the number of (stacked) nucleosomes contributing to the array folding ($N_{fold}$). Three other parameters are also calculated: the fibre stiffness ($k_{fibre}$, stage 1 in Fig. 4A,B), the free energy difference between the folded fibre and an extended beads-on-a-string conformation ($\Delta G_1$, stage 2 in Fig. 4A,B), and the free energy cost of the DNA unpeeling up to one DNA turn on the histone octamer ($\Delta G_2$, stage 3 in Fig. 4A,B).

We noted that TRF2$^{\Delta N}$ presence at a concentration as low as 1 nM results in a noticeable change in the force–extension curve of the #Telo-18# arrays. Figure 4C compares typical stretching curves obtained for the #Telo-18# array in the presence of 1 nM TRF2$^{\Delta N}$ in the measurement buffer with results in the absence of the dimer. Considering the general appearance of the stretching curve in the presence of TRF2$^{\Delta N}$, it is striking that the folded fibre becomes

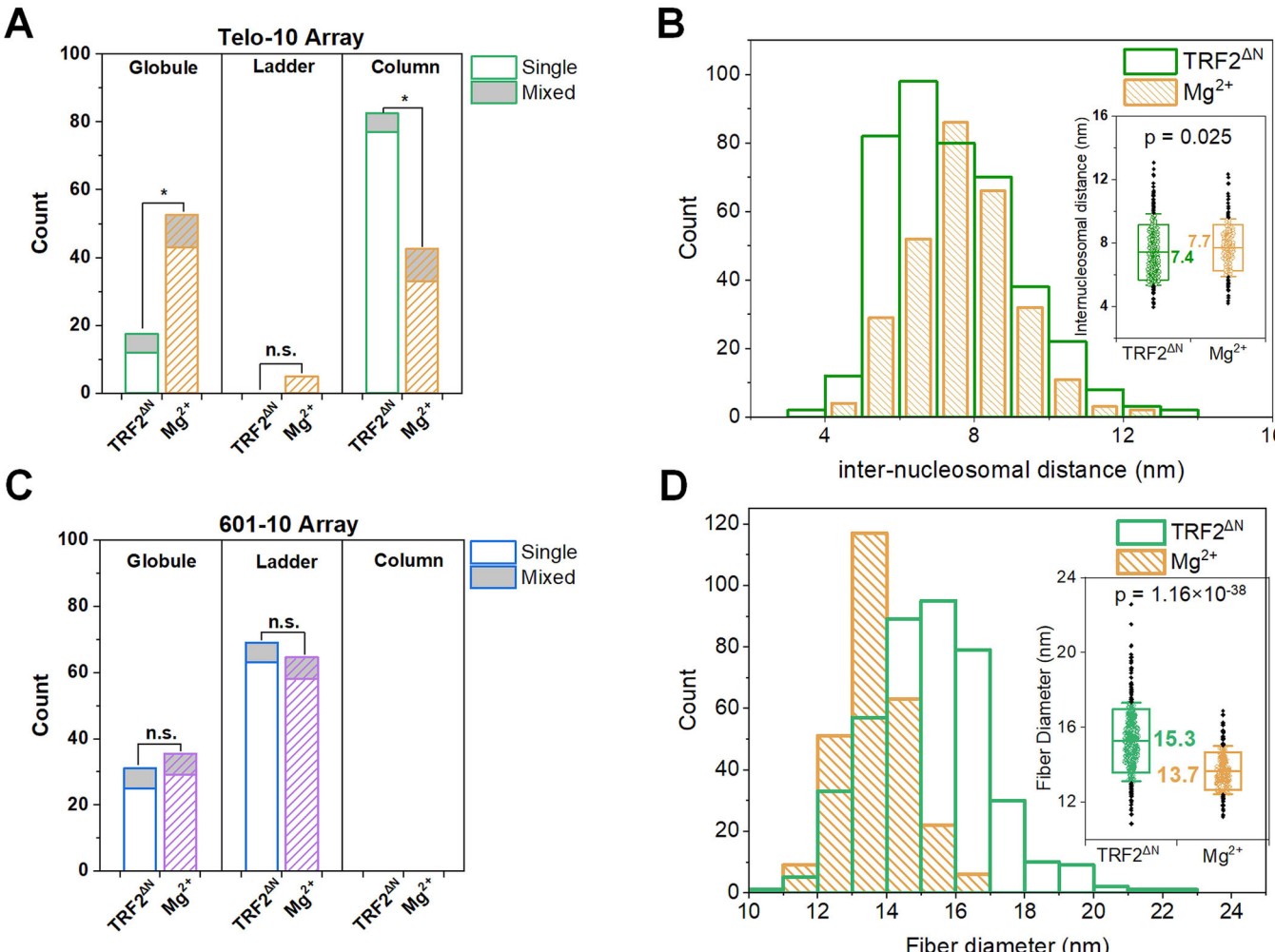

**Figure 3. Characterisation of telomeric and 601 array compaction.**

(A) The structural features of compacted Telo-10 fibres were determined from negative-stained EM micrographs in the presence of TRF2$^{\Delta N}$ dimer (green) and Mg$^{2+}$ (orange). 100 single and compacted fibres from each sample (n = 3, technical replicates) were categorised into three structure classes: globule, ladder, and column. The difference in structural features of fibres in the presence of TRF2$^{\Delta N}$ or Mg$^{2+}$ was compared by a two-sample proportion test; a p value < 0.05 was considered significant (marked with *), while a p value ≥ 0.05 was considered not significant (marked with n.s.). The p values for the structural difference in globule, ladder and column features were 2.15×10$^{-8}$, 0.054 and 5.64×10$^{-9}$, respectively. (B) Histogram plot for the inter-nucleosomal distance measured from Telo-10 array in the presence of TRF2$^{\Delta N}$ (green) or Mg$^{2+}$ (orange). The insert shows the box plots overlayed with the data points (diamond symbols), with statistical analysis performed using a two-sample t-test with equal variance; p value = 0.025. Mean values are indicated in the insert; boxes show mean ± s.d.; whiskers indicate the 10-90% data range; black diamond symbols are outliers' data points. For Telo-10 array in the presence of TRF2$^{\Delta N}$: n = 417, mean = 7.4 nm, s.d. = 1.7 nm and range = 3.9–13.1 nm. For Telo-10 array in the presence of Mg$^{2+}$: n = 285, mean = 7.7 nm, s.d. = 1.5 nm and range = 4.2–12.3 nm. The data presented are from three technically replicated experiments. (C) The structural features of compacted 601-10 fibres were determined from negative-stained EM micrographs in the presence of TRF2$^{\Delta N}$ dimer (blue) and Mg$^{2+}$ (purple). 100 single and compacted fibres from each sample (n = 3, technical replicates) were categorised into three structure classes: globule, ladder, and column. The difference in structural features of fibres in the presence of TRF2$^{\Delta N}$ or Mg$^{2+}$ was compared by a two-sample proportion test; a p value < 0.05 was considered significant (marked with *), while a p value ≥ 0.05 was considered not significant (marked with n.s.). The p values for the structural difference in globule and ladder features were 0.55 and 0.45, respectively. (D) Histogram plot for fibre diameter measured from Telo-10 array in the presence of TRF2$^{\Delta N}$ (green) or Mg$^{2+}$ (orange). The insert shows the box plots overlayed with the data points (diamond symbols), with statistical analysis performed using a two-sample t test with equal variance; p value = 1.16 × 10$^{-38}$. Mean values are indicated in the insert; boxes show mean ± s.d.; whiskers indicate the 10–90% data range; black diamond symbols are outliers' data points. For Telo-10 array in the presence of TRF2$^{\Delta N}$: n = 412, mean = 15.3 nm, s.d. = 1.7 nm and range = 10.8–15.2 nm. For Telo-10 array in the presence of Mg$^{2+}$: n = 268, mean = 13.7 nm, s.d. = 1.0 nm and range = 11.2–13.6 nm. The data presented are from three technically replicated experiments. Source data are available online for this figure.

noticeably stiffer; the plateau region corresponding to the fibre unfolding and partial DNA unwrapping is shifted to larger force, and the high-force region (>10 pN) does not show the apparent stepwise pattern observed for practically all previously studied nucleosome arrays (see, e.g. (Brower-Toland et al, 2002; McCauley

et al, 2022; Meng et al, 2015). The same appearance is manifested at TRF2$^{\Delta N}$ concentrations of 10 nM and 100 nM (Fig. 4D,E). The results show similar force-extension behaviour compared to the data obtained at 1 nM TRF2$^{\Delta N}$, and the general display of the curves implies significantly stiffer fibres that exhibit considerable

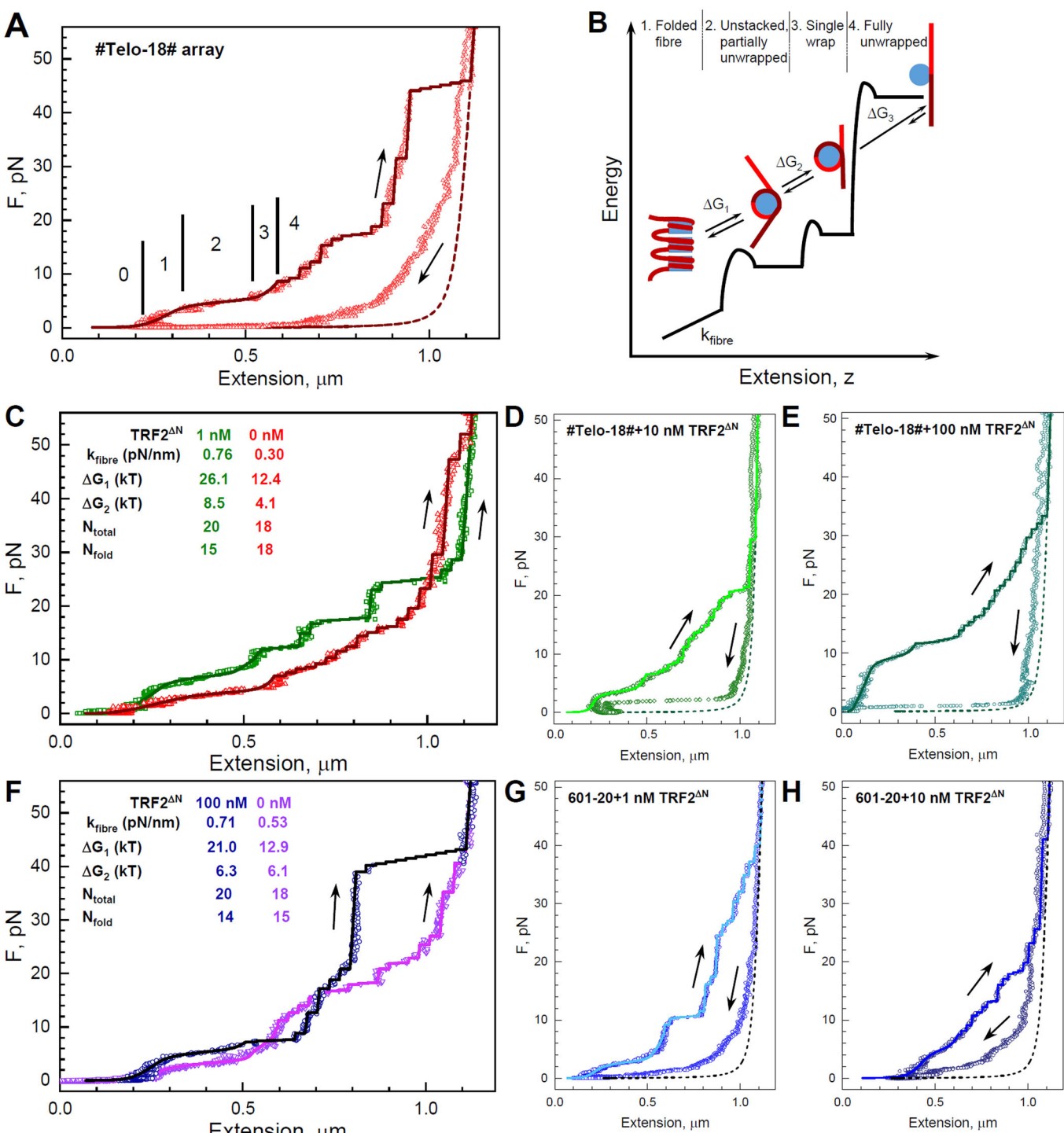

resistance to force upon TRF2$^{\Delta N}$ binding. This suggests that the major qualitative change of the mechanical properties of the arrays occurred already at 1 nM concentration of the TRF2$^{\Delta N}$ dimer, but the effect on the mechanical properties of the telomeric chromatin fibre is amplified when more TRF2$^{\Delta N}$ dimers are bound.

The results reflect the high affinity of the TRF2$^{\Delta N}$ for the telomeric chromatin. It has to be noted that even at 1 nM, TRF2$^{\Delta N}$ dimers are in large excess over nucleosome arrays since the

concentration of the fibres attached to the magnetic beads and the glass surface is extremely low (~30 ng/µl stock solution was diluted 40,000–50,000 times in the process of the fibre-ends attachment followed by further depletion during washing of the flow cell). It is reasonable to assume that the fibre binds a substantial amount of TRF2$^{\Delta N}$ dimers in the 1 nM solution, and further increase of the protein concentration leads only to moderate additional binding of the TRF2$^{\Delta N}$.

**Figure 4. Single-molecule force spectroscopy of the #Telo-18# and 601-20 nucleosome arrays with and without TRF2$^{\Delta N}$ dimer.**

(A) An example of an experimental stretch-relief curve of the #Telo-18# nucleosome array recorded in the absence of TRF2$^{\Delta N}$. Different stages of the fibre stretching are indicated with numbers referring to the respective transition according to the model shown in (B). In (A, C–H), points are experimental data; solid lines show the model fitting; dashed lines are force-extension dependence of bare 3242 bp DNA calculated using the worm-like chain (WLC) approximation. (B) Statistical mechanics model for the single-molecule nucleosome array stretching. Free energy – extension scheme illustrating different stages of the nucleosome array transformation under the influence of the stretching force. Deformation of the fibre includes (0) extension of the bare DNA; (1) extension of the folded array characterised by stretching modulus, $k_{fibre}$; (2) transition of the array from a folded fibre to a beads-on-a-string chain accompanied by nucleosome unstacking and partial DNA unwinding as quantified by the number of folded nucleosomes ($N_{fold}$) and free energy change ($\Delta G_1$); (3) deformation of the nucleosomes with further DNA unpeeling ($\Delta G_2$); (4) one-step rupture of the last turn of the DNA wrapped on the histone core observed for all nucleosomes in the array ($N_{total}$). (C–E) Examples of force spectroscopy data obtained for the #Telo-18# arrays in the presence of 1 nM (C), 10 nM (D), and 100 nM (E) TRF2$^{\Delta N}$ dimer. Points are experimental data; lines are fittings to the statistical mechanics model using an NRL equal to 157 bp. In (C), the stretching curve of the #Telo-18# + 1 mM TRF2$^{\Delta N}$ system (green) is compared with the data recorded for the array in the TRF2$^{\Delta N}$ absence (red). (F–H) Examples of force spectroscopy data obtained for the 601-20 arrays in the presence of 100 nM (F), 1 nM (G), and 10 nM (H) TRF2$^{\Delta N}$ dimer. Points are experimental data; lines are fittings to the statistical mechanics model with 157-bp NRL. In (F), the stretching curve of the 601-20 + 100 mM TRF2$^{\Delta N}$ system (dark blue) is compared with the data recorded for the array in the TRF2$^{\Delta N}$ absence (blue). Source data are available online for this figure.

In contrast, the addition of the TRF2$^{\Delta N}$ dimer to the flow cell does not significantly influence the 601-20 arrays, even in the presence of 100 nM protein concentration (Fig. 4F–H). This is in remarkable contrast to the #Telo-18# arrays data, which demonstrates a noticeable increase in resistance to the applied stretching force already in the presence of 1 nM TRF2$^{\Delta N}$ dimer. The different response of the #Telo-18# and 601-20 arrays to the addition of the TRF2$^{\Delta N}$ dimer is not always seen from individual stretching curves, which show significant variations. However, it becomes apparent upon comparison of the parameters calculated from the statistical mechanics model. This data is summarised in Fig. 5A–E (compare left- and right-hand graphs of each panel of this figure; numerical values are listed in the Appendix Table S1).

We suggest that the TRF2$^{\Delta N}$ protein binds to the non-telomeric 601-20 arrays in a non-specific manner driven by primarily electrostatic interactions. This binding mode produces a slight increase of the $N_{total}$, $N_{fold}$, $\Delta G_1$, and $k_{fibre}$ values observed at 100 nM TRF2$^{\Delta N}$ dimer, while lower TRF2$^{\Delta N}$ concentrations (1 nM and 10 nM) produce a weaker influence on the mechanical properties of the 601-20 arrays (Figs. 4G,H and 5A–E, right-hand panels). Earlier force spectroscopy studies have shown that such interaction is accompanied by changes in the force–extension curves similar to the effects observed in our earlier 601-20 array MMT data (Soman et al, 2022b).

In addition, the relaxation half of the stretch-relief curve shows deviation from the typical behaviour of both bare DNA (dashed lines in Figs. 4A,D,E and EV4) and the #Telo-18# array in the absence of TRF2$^{\Delta N}$ (Fig. 4A). At low force (in the range of 1-4 pN), the fibre abruptly contracted from an extension of about 0.8 μm to the length roughly corresponding to its initial dimension. The relaxation behaviour of the fibre is similar to that observed in recent magnetic tweezer experiments studying TRF2 interactions with telomeric, λ-phage, and mixed-sequence DNAs (Soranno et al, 2022). In the cited work, the presence of 300 nM full-length TRF2 resulted in highly heterogeneous stretching and relaxation behaviour under the influence of mechanical force in the range of 0.5–6 pN. In our MMT experiment, the application of a high stretching force (56 pN) resulted in substantial histone dissociation from the DNA and the subsequent relaxation behaviour due to force decrease is expected to act on nucleosome-depleted DNA with bound TRF2$^{\Delta N}$.

Most MMT data can be reliably fitted to the current statistical mechanics model used previously to describe nucleosome array stretching (Brouwer et al, 2021; Kaczmarczyk, 2019; Korolev et al,

2022; Meng et al, 2015). In the absence of any nucleosome positioning signal in the telomeric DNA sequence, independent methods are necessary to determine an NRL value required to fit MMT data for the #Telo-18# arrays. In this work, we use NRL = 157 bp, assuming that fibre adopts the NRL value directed by the two flanking '601' sequences in the #Telo-18# DNA construct. This NRL also allows direct comparison with the MMT data collected for the 601-20 arrays reconstituted on the positioning DNA sequence with 157 bp NRL. All arrays show a wide distribution of parameter values, but the MMT method allows the collection of many traces that enables statistically significant differences to be ascertained.

An increase in TRF2$^{\Delta N}$ concentration leads to the stabilisation of the nucleosomes in the #Telo-18# arrays. The total number of nucleosomes increases with TRF2$^{\Delta N}$ concentration (Fig. 5A), which suggests that TRF2$^{\Delta N}$ binding stabilises the nucleosomes on the telomeric DNA, which lacks high-affinity nucleosome positioning properties. The number of (stacked) folded nucleosomes remains similar (Fig. 5B). Upon TRF2$^{\Delta N}$ binding, the #Telo-18# arrays display considerably higher $\Delta G_1$ and $\Delta G_2$ values (Fig. 5C,D, respectively) and become stiffer (larger $k_{fibre}$ value, Fig. 5E).

A small fraction of the #Telo-18# and 601-20 fibres display stretching profiles, which cannot be fitted using the current statistical mechanics' approach (Fig. EV4). These profiles are characterised by a shift to higher force and loss of clear separation between different stages of fibre unwinding and DNA-histone core unpeeling. We suggest that this behaviour might correspond to a fraction of the highly compact and heterogeneous population of the TRF2$^{\Delta N}$ dimer-nucleosome array complexes.

Next, we analysed rupture events corresponding to the largely irreversible DNA unpeeling from the histone core at the high force (stage 4 in Fig. 4A,B). Distributions of rupture forces, $F_{rupt}$, as a function of the pulling rate ($dF/dt$) and the number of the nucleosome in the array ($N$) in the form $\ln[(N_0 - N + 1/N)\cdot dF/dt]$ observed for the #Telo-18# and 601-20 arrays in the absence and presence of the TRF2$^{\Delta N}$ dimer are shown in Fig. EV5. Plotting $F_{rupt}$ in these coordinates allows linear fitting to determine the rate constant for nucleosome disruption under zero external force ($k_{off}$, Fig. 5F, left graph) and the distance between the bound state and the activation barrier peak ($d$, Fig. 5F, right graph) (Pope et al, 2005). To ensure the robust fit, the impact of outliers was reduced by including only data points in bins that exceeded a threshold in the 2D histogram of 0.5 times the peak value (indicated as the

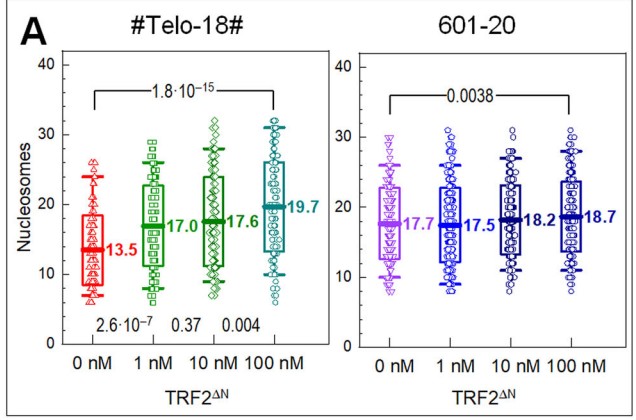

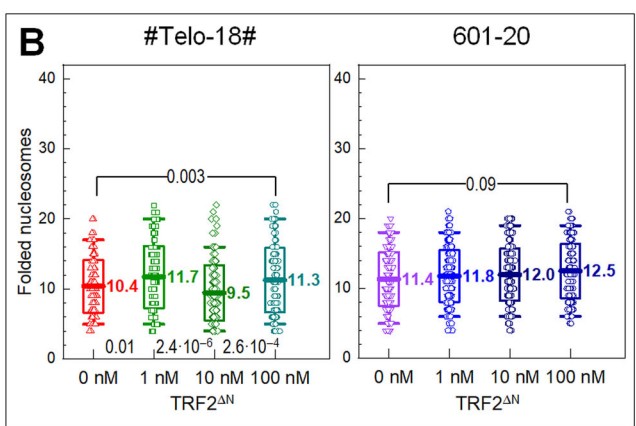

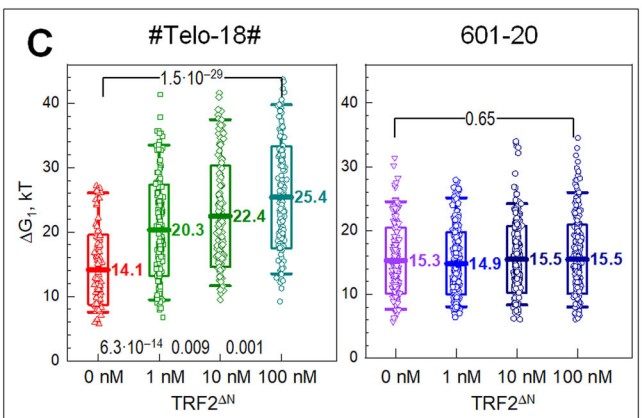

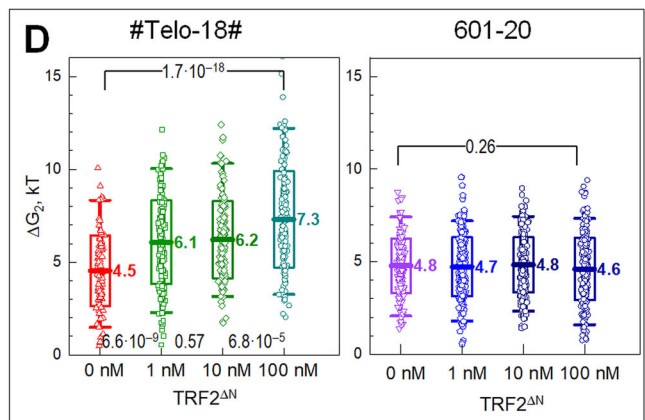

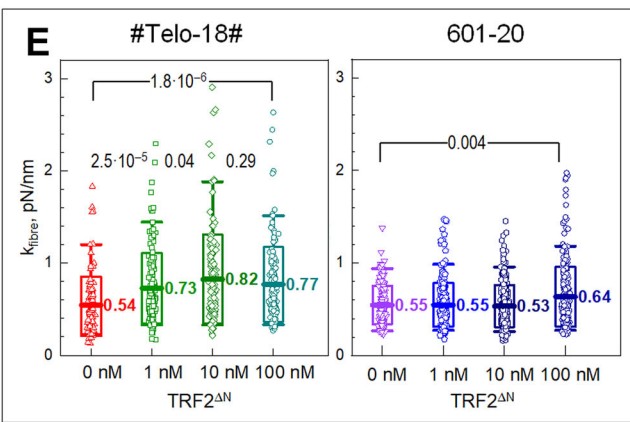

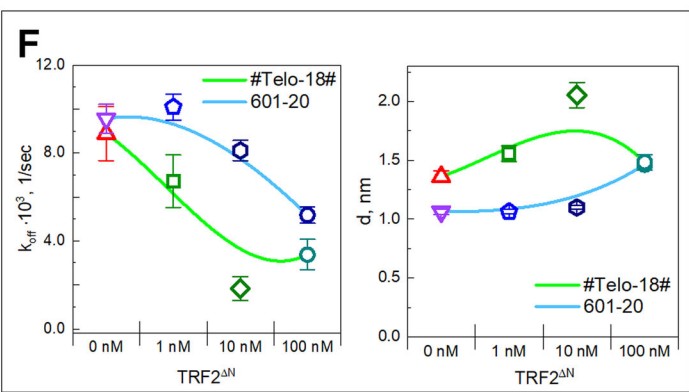

coloured contours in Fig. EV5). Application of this cut-off discards about 50% of the datapoints. Comparison with previously published data sets for the #Telo-18# and 601-20 arrays without TRF2$^{\Delta N}$ (Soman et al, 2022b) yielded only a small difference in the rupture characteristics.

For the #Telo-18#, linear fittings of the $F_{rupt}$ data points in the range 7 pN < $F_{rupt}$ < 30 pN yield higher values and lower slopes for the arrays in the presence of TRF2$^{\Delta N}$ (Fig. EV5, left- column graphs; Appendix Table S2). The strength of the DNA-histone binding is consequently stronger in the presence of TRF2$^{\Delta N}$ compared to its absence. This is reflected by the lower $k_{off}$ values (Fig. 5F, left graph), indicative of the resistance of the DNA-histone attachment to the applied force. Furthermore, TRF2$^{\Delta N}$ binding leads to an

increase in the *d* values (Fig. 5F, right graph). i.e. a shift of the activation barrier for the rupture event in the presence of TRF2$^{\Delta N}$. In contrast, the rupture forces observed for the 601-20 arrays do not show a noticeable influence of the TRF2$^{\Delta N}$ addition (Fig. EV5, right-hand panels and Fig. 5F).

The parameters $k_{off}$ and *d* for the #Telo-18# and 601-20 arrays (calculated from the linear fittings of the $F_{rupt}$ versus *x*) are displayed in Fig. 5F. For the #Telo-18# arrays, presence of TRF2$^{\Delta N}$ increases the rupture length *d* from about 1.1 to up to 2.1 nm (at 10 nM TRF2$^{\Delta N}$) and decreases the rupture rate $k_{off}$, from about $8.9 \times 10^{-3}$ down to $1.8 \times 10^{-3}$ s$^{-1}$ (at 10 nM TRF2$^{\Delta N}$). The rupture length decreased at 100 nM TRF2$^{\Delta N}$, possibly by the effect of multiple TRF2$^{\Delta N}$ protein binding to the telomeric nucleosome.

**Figure 5. Influence of the TRF2$^{\Delta N}$ dimer on mechanical properties of the #Telo-18# and 601-20 arrays.**

(A–F) Present the summary of the parameters calculated from the force spectroscopy data of the #Telo-18# and 601-20 arrays determined in the presence (0 nM) and the absence of the 1 nM, 10 nM, and 100 nM of the TRF2$^{\Delta N}$ dimer. In these panels, the left-hand graphs are the results for the #Telo-18#, and the right-hand graphs are for the 601-20 array data. The experimental value for each trace is shown as a symbol; mean values are indicated in the graphs; boxes display mean value ± standard deviation; whiskers indicate the 5–95% data range. For the #Telo-18# arrays, the $p$ values are calculated by $t$ test, assuming equal variance for the respective pairs, and are displayed at the bottom of the graphs. In (A–E), $p$ values between parameters determined at 0 nM and 100 nM TRF2$^{\Delta N}$ dimer are displayed at the top of the graphs. Numerical data (including mean ± s.d. and numbers of measured points, $n$) is given in Appendix Table S1. (A) Number of nucleosomes ($N_{total}$) in the #Telo-18# (left panel) and 601-20 arrays (right panel). (B) Number of folded nucleosomes ($N_{fold}$) in the #Telo-18# (left panel) and 601-20 arrays (right panel). (C) Free energy of the nucleosome array unfolding and partial unfolding ($\Delta G_1$). Data for the #Telo-18# (left panel) and 601-20 arrays (right panel). (D) Free energy of unpeeling the 13 bp DNA from the histone core ($\Delta G_2$). Data for the #Telo-18# (left panel) and 601-20 arrays (right panel). (E) Stiffness of the folded nucleosome fibre ($k_{fibre}$). Data for the #Telo-18# (left panel) and 601-20 arrays (right panel). (F) Parameters ($k_{off}$) (left panel; rate constant for the nucleosome disruption under zero external force) and $d$ (right panel; the distance between the bound state and the activation barrier peak along the direction of the applied force) of the rupture transition calculated by linear fitting of the equation above for the #Telo-18# and 601-20 arrays. Values of the $d$ and $k_{off}$ were determined from the slope and intercept of the linear fit in coordinates $F_{rupt}$ versus $\ln\left[\left(N_0 - N - 1/N\right)\frac{dF}{dt}\right]$ (see Fig. EV5). Lines are drawn to show the trend of the data. Numerical values (including mean ± s.d. and numbers of measured points, $n$) are given in Appendix Table S2. Source data are available online for this figure.

Nucleosomes in the 601-20 array were less affected by the TRF2$^{\Delta N}$ addition. Only at 100 nM TRF2$^{\Delta N}$, both the rupture length and the rupture rate of the 601 nucleosomes do change and approach the level of the telomeric nucleosomes (see Fig. 5F and Appendix Table S2). Overall, analysis of the rupture forces showed that TRF2$^{\Delta N}$ has a stabilising effect on nucleosomes, which is more pronounced on the telomeric than on the 601 nucleosomes. Our $d$ and $k_{off}$ values are in the range of the earlier published results determined under significantly different settings (different solution conditions, dissimilar NRL of the arrays, optical versus magnetic tweezers) (McCauley et al, 2018; McCauley et al, 2022). In these studies, the addition of the nucleosome-destabilising proteins (HMGB or FACT) leads to the expected increase of the $k_{off}$ value (named opening rate in the above-cited papers). In contrast, in our work, the stabilising influence of the TRF2$^{\Delta N}$ protein is reflected by the decrease of $k_{off}$, indicating an increase in nucleosome stability.

In conclusion, the force spectroscopy stretching analysis using the MMT method shows that TRF2$^{\Delta N}$ dimer specific binding to the columnar telomeric chromatin fibres results in significant changes in the mechanical properties of the #Telo-18# fibres and the thermodynamic parameters that characterise the DNA unwrapping during the different stages of stretching. The fact that TRF2$^{\Delta N}$-bound columnar fibres are stiffer is in agreement with a model where the TRF2$^{\Delta N}$ binding was proposed to be located in the DNA supergroove as previously suggested (Soman et al, 2022b) and extends between different neighbouring nucleosomes. Such a model would explain a stiffer fibre and the higher energy cost related to DNA unwrapping, characterised by the free energies $\Delta G_1$ (stage 2) and $\Delta G_2$, stage 3 (see Fig. 4A,B) and the parameters describing the rupture events. Remarkably, the arrays reconstituted on the 157 bp-NRL nucleosome positioning sequence do not show such a pronounced influence of the TRF2$^{\Delta N}$ binding on the mechanical properties of the 601-20 arrays. The MMT data are also consistent with the EM and AUC-SV results, demonstrating the presence of the columnar compaction upon TRF2$^{\Delta N}$ binding and the effect of the TRF2$^{\Delta N}$ dimer association on the average columnar fibre width.

## Discussion

In addition to its specific functions at the telomeres, TRF2 performs various roles in other parts of chromatin (Baker et al, 2011; Imran

et al, 2021; Mukherjee et al, 2019). Our study confirms that TRF2$^{\Delta N}$ dimers bind both telomeric and non-telomeric nucleosome arrays. The addition of the TRF2$^{\Delta N}$ or full-length TRF2 dimers to the nucleosome arrays causes array compaction. However, TRF2 binding makes distinctly different effects on the folded compact structure of the telomeric and '601' arrays: The TRF2 dimers organise telomeric arrays into regular columnar structures, whereas the protein binding to the '601' array results in the formation of ladder-like architecture.

In our previous work (Soman et al, 2022b), we revealed that the presence of Mg$^{2+}$ leads to condensed telomeric chromatin and that these fibres are organised into a columnar structure. We also determined the cryo-EM structure of the telomeric tetranucleosome. In this study, we show that the presence of the TRF2$^{\Delta N}$ or TRF2 protein alone was sufficient to induce the columnar architecture of the Telo-10 fibres more efficiently than Mg$^{2+}$, as inferred from the observation of a significantly larger fraction of columnar fibres in the presence of the TRF2 protein. TRF2$^{\Delta N}$ compacted telomeric fibres are also more compact. This shows that the TRF2 protein binding mediates the organisation of compact telomeric chromatin. We tentatively propose that the TRF2 dimer binds to the DNA 'supergrooves' and enables the closely stacked columnar fibre organisation. Our AUC-SV and negative stain EM experiments showed that TRF2 induced homogeneous compaction of the telomeric chromatin. The observed decrease in the FWHM of the AUC-SV peaks, combined with the absence of peaks corresponding to oligomers of the array, suggests intra-array compaction mediated by TRF2$^{\Delta N}$ interaction with adjacent nucleosomes within the array resulted in the columnar stacking of adjacent nucleosomes. Recent observation of native chromatin at telomeres showed that the fibre width, when measured by tagging with TRF2 (15.4 nm), was approximately 1 nm larger than the fibre width measured by tagging the H2B histone (14.4 nm) (Hübner et al, 2022). This suggests that TRF2 is most likely bound on the surface of the telomeric chromatin, increasing the fibre width. Our results showed a similar mean fibre diameter of 15.3 nm in the presence of the TRF2$^{\Delta N}$ protein, 1.6 nm wider than Mg$^{2+}$-compacted Telo-10 fibre (13.7 nm) (Fig. 3D). The intra-array compaction and the proposed binding of TRF2 to the supergrooves on the surface of chromatin, coupled with the fact that TRF2 binds as a dimer, suggests that the two monomers of the TRF2 dimer bind on the surface of the telomeric nucleosome array. The supergrooves

we identified in the compact telomeric chromatin structure provide an ideal binding site for the TRF2 dimer and is consistent with the columnar conformation observed upon the binding of TRF2 to telomeric chromatin. We hypothesise that the TRF2 dimer engages the aligned major and minor grooves spanning multiple nucleosomes in its interaction with telomeric chromatin. The binding of TRF2 to supergrooves is expected to lead to columnar stacking and subsequent stabilisation by the histone tails elements identified in our recent cryo-EM study (Soman et al, 2022b). This model agrees with the fibre thickness observed in native telomeric chromatin ~14 nm. These results and this model are also consistent with our single molecule magnetic tweezer stretching experiments, which displayed profound changes in the mechanical (stiffer fibres) and thermodynamic properties characterising the energy cost upon DNA unwrapping, demonstrating stabilisation of the fibres in the presence of TRF2$^{\Delta N}$. Such telomere stabilisation in the presence of TRF2 may be essential for capping the chromosome ends and could hence be important in telomere maintenance.

The TRF2 protein has been reported to promote the assembly of T-loop structure in telomeres in vivo and in vitro (Doksani et al, 2013; Stansel et al, 2001; Timashev and de Lange, 2020). As such, it is possible that dimerisation of TRF2 binding to distal nucleosomes can potentially promote the formation of a T-loop structure or even bridge distal nucleosome fibres, forming a network of stable structures to protect the chromosome ends. To establish the molecular details of TRF2-telomeric chromatin interactions, studies aimed at characterising the high-resolution structure of TRF2-telomeric chromatin fibre complexes are the next step in elucidating the details of TRF2 and other shelterin components interacting with the telomere.

# Methods

## Preparation of the nucleosome arrays

Telo-10 and 601-10 DNA templates were amplified and purified as described previously (Soman et al, 2022b). Briefly, both DNA templates were multimerised to comprise 10 repeats of 157 bp units, interspaced with the AvaI recognition site, and were purified by size exclusion chromatography. The telomeric tetranucleosome DNA template (Telo-4) was purified employing PEG fractionation followed by ion-exchange chromatography (Soman et al, 2022b).

A hybrid telomeric/601 DNA (#Telo-18#) consisting of $18 \times 157$ bp telomeric DNA units flanked by a 157 bp Widom 601 DNA fragment at each terminus (labelled as #) template was also prepared for the studies of the nucleosome arrays by multiplexed magnetic tweezers (MMT) measurements. The #Telo18# template was synthesised by large-scale ligation of #Telo-8 and Telo-10# DNA sequences as described in (Soman et al, 2022b). #Telo-8 and Telo-10# DNA arrays were built by extending the upstream side of a Telo-8 array and the downstream side of a Telo-10 array with a $1 \times 157$ bp Widom 601 insert. End-labelling of the #Telo18# template by digoxigenin and biotin for the MMT measurements was performed according to the published procedure (Soman et al, 2022b).

The recombinant human histone proteins were individually expressed, purified, and refolded into human histone octamers, as previously described (Dyer et al, 2004; Luger et al, 1999). Briefly,

individual histone proteins (H2A, H2B, H3, and H4) were mixed under high salt at a ratio of 1.2:1.2:1.0:1.0, and the assembled octamer was purified by size-exclusion chromatography.

The nucleosome arrays for the EMSA, AUC, EM, and MMT studies were assembled in vitro using respective DNA templates and recombinant human histone octamer by salt dialysis (Huynh et al, 2005; Soman et al, 2022b). For each reconstitution of nucleosome arrays, the stoichiometric amount of histone octamers was empirically determined by titrating the different amounts of histone octamer:DNA ratios (Appendix Fig. S1G–I). During salt dialysis, the salt concentration was reduced from 2 M to 2.5 mM LiCl by continuous dialysis at a 0.6 ml/min flow rate overnight using a peristaltic pump. The final buffer consisted of 10 mM HEPES-LiOH pH 7.4, 1 mM Li-EDTA, 1 mM DTT, and 2.5 mM LiCl. The saturation level of the reconstituted arrays was assessed on 0.7% agarose EMSA ran in 0.2×TB-agarose gel buffer, post-stained with 0.2×SYBR™ Gold Nucleic Acid Gel Stain (Thermo-Fisher Scientific, USA).

Generally, EMSA, AUC-SV, EM, and MMT studies of the TRF2$^{\Delta N}$ binding to nucleosome arrays were carried out with nucleosome arrays at and/or near the saturation point (black triangles in Appendix Fig. S1G–I). The saturation histone octamer:DNA ratio was first established in a series of small-scale array reconstitutions and further fine-tuned with narrow titration of histone octamer in large-scale array reconstitution.

## Expression and purification of TRF2$^{\Delta N}$ and TRF2

The His-tagged TRF2$^{\Delta N}$ protein (residue 42–542, with a deletion of Alanine at the 477 position) encoded in pET30a was a kind gift from Prof. Daniela Rhodes. The protein was expressed and purified as described previously (Fairall et al, 2001). Briefly, the His-tagged TRF2$^{\Delta N}$ was expressed with isopropyl-β-D-thiogalactopyranoside induction and purified using Ni-NTA resin and size-exclusion chromatography. The His-tag was cleaved with TEV protease before the purification by size-exclusion chromatography.

An expression-optimised *E. coli* (BL21 (DE3) Rosetta T1R) clone for FL TRF2 encoded in a pNH-Trxt vector was purchased from the Protein Production Platform (PPP) at Nanyang Technological University. The expression and purification of TRF2 were carried out identically to the TRF2$^{\Delta N}$.

## Electrophoretic mobility shift assay

An assay of the TRF2$^{\Delta N}$ binding to nucleosome array was performed at 33.7 nM 10-mer array concentration in the range 0–1.35 µM of TRF2$^{\Delta N}$ dimer (0-4.0 TRF2$^{\Delta N}$ dimer/157 bp DNA ratio) in 20 mM HEPES pH 7.4, 5 mM LiCl, 0.075 mM EDTA, and 0.75 mM DTT. The samples were incubated at 4 °C for 1 h before being fractionated on 0.7% TB-agarose gel, running with 0.2×TB buffer at 150 V for 60 min. The agarose gel was post-stained with 0.2×SYBR Gold Nucleic Acid Gel Stain for 40 min for imaging.

## Precipitation assay

TRF2$^{\Delta N}$ binding to Telo-10 arrays was prepared as mentioned above in a buffer of 20 mM HEPES, 5 mM LiCl, 0.075 mM EDTA, and 0.75 mM DTT. The TRF2$^{\Delta N}$-array samples were added with MgCl$_2$ to a final concentration of 5 mM and further incubated at

room temperature for 15 min. After which, the samples were centrifuged at $20,000 \times g$ for 15 mins at 4 °C. The supernatant and pellet fractions were separately analysed on an 18% SDS-PAGE.

## Analytical ultracentrifugation (AUC)

TRF2$^{\Delta N}$ binding to nucleosome arrays and DNA was prepared as mentioned above in a buffer of 20 mM HEPES, 5 mM LiCl, 0.075 mM EDTA, and 0.75 mM DTT. Mg-compacted nucleosome arrays were prepared in a buffer of 2.5 mM LiCl, 10 mM HEPES (pH 7.4), 0.1 mM EDTA, 1 mM DTT and 0.0-0.2 mM MgCl$_2$. All samples used for the AUC-SV study had an optical density of $A_{259} = 0.6$–0.8 cm$^{-1}$. AUC-SV experiments were conducted using a Beckman XL-I analytical ultracentrifuge equipped with an eight-channel AN-50 Ti analytical rotor and a Beckman monochromator. The analytical sample cell was assembled with 2-sector Epon (epoxy) centrepieces with a path length of 12 mm.

For nucleosome array samples containing TRF2$^{\Delta N}$, the samples were loaded into pre-chilled cells and rotor before equilibrating under vacuum at 4 °C for 2 h, followed by a single scan radial calibration at 3000 rpm. Then, 85 scans of absorbance data were collected at 12,000 rpm at 10-min intervals using ProteomeLab XL-I software. For nucleosome array samples incubated with Mg$^{2+}$, the cells containing the samples were loaded into the rotor and equilibrated under vacuum at 20 °C for 2 h, followed by a single scan radial calibration performed at 3000 rpm. Then, 60 scans of absorbance data were collected at 12,000 rpm at 10-min intervals using ProteomeLab XL-I software. For DNA samples containing TRF2$^{\Delta N}$, the cells were loaded into the rotor and equilibrated under vacuum at 20 °C for 2 h, followed by a single scan radial calibration performed at 3000 rpm. Then, 60 scans of absorbance data were collected at 30,000 rpm at 10-min intervals using ProteomeLab XL-I software. Three experimental repeats were done to obtain an average $s_{20,w}$.

The AUC-SV data were analysed with the SEDFIT program using a continuous c(s) distribution model (Schuck, 2000). The analysis was refined at a resolution of 200, and the sedimentation coefficient was corrected to $s_{20,w}$ using a partial specific volume calculated from the DNA and proteins mass ratio: 0.556 ml/g for DNA; 0.662 and 0.6655 ml/g for the 10-mer array (at 4 °C and 20 °C, respectively); 0.677–0.691 ml/g for the 10-mer array with added TRF2$^{\Delta N}$ at 20 °C. The density and viscosity were adjusted according to the buffer composition and analysis condition calculated from the Sednterp program (Mukherjee et al, 2019).

## Negative-stain electron microscopy

The nucleosome arrays were buffer exchanged to 20 mM potassium cacodylate and 1 mM DTT and mixed with 0–0.046 µM TRF2$^{\Delta N}$ dimer (corresponding to 0–0.4 TRF2$^{\Delta N}$ dimer/157 bp DNA ratios) at a final array concentration of 12 nM. The samples were incubated at 4 °C for 1 h before applying them to the grid. For Mg-compacted nucleosome arrays, 12 nM of the arrays were incubated in 20 mM potassium cacodylate and 1 mM DTT with the indicated concentration of MgCl$_2$ for 16 h at 4 °C before applying it to the grid.

The 10 nm carbon-coated copper grid (400 square mesh) was used for negative-stain EM. First, the grid was glow-discharged for 1 min. Next, 5 µl of the array sample was deposited on the grid and

incubated for 1 min. The excess array sample was blotted away using a Whatman® qualitative filter paper. The grid was stained with 2% uranyl acetate for an additional minute before the excess stain was removed by blotting. The grids were visualised with a Tecnai T12 electron microscope operating at 120 kV, and micrographs were recorded on FEI Eagle 4 K CCD camera at ×49,000 magnification using Thermofisher TEM imaging and analysis (TIA) software.

The micrographs were analysed on FIJI/IMAGEJ program (Schneider et al, 2012), and the results were plotted with OriginPro software (OriginPro, 2019). To quantify fibre features, 100 fibre particles from each sample were categorised into three structure classes: globule, ladder, and column. A scoring system was adopted to accurately account for the fibres' features, as some fibres displayed two distinct features. A score of 1 was given to particles with a single structural feature, assigning a score of 0.5 to the arrays displaying two structural features in each particle. The $p$ value for structural features was calculated by OriginPro software using a two-sample proportion test. For inter-nucleosomal distance, the mid-point of two nucleosomes within the fibre was measured to obtain a mean value. For fibre diameter, the width of a trinucleosome within the fibre was measured to obtain a mean value: two parallel lines were used to demarcate the edge of the trinucleosome, and the perpendicular distance was measured. The parallel lines were adjusted to fit the next trinucleosome within the fibre. The $p$ values for inter-nucleosomal distance and fibre diameter were calculated by OriginPro software using a two-sample $t$ test assuming equal variance.

## Cryo-electron microscopy

The reconstituted nucleosome array and TRF2/TRF2$^{\Delta N}$ were buffer-exchanged to 20 mM potassium cacodylate and 1 mM DTT. The arrays and TRF2 were mixed to achieve the desired TRF2/NCP ratio at a final 1 mg/ml concentration of the array and stored overnight at 4 C. Four µl of the mixture was applied to glow discharged Quantifoil R1.2/1.3 200 mesh holey carbon grids. The grids were blotted for 1.5 s with a blot force of 1 and plunge frozen in liquid ethane at 93 K, employing Vitrobot Mark IV (Thermofischer scientific). The grids were visualised on Tecnai Arctica (200 kV) (Thermofischer scientific) equipped with Falcon 3EC director electron detector (4 K × 4 K pixels).

## Multiplexed Magnetic Tweezers (MMT) measurements

The homemade MMT setup and flow cells used in this work were assembled at Nanyang Technological University using the design developed in the laboratory of Chromatin Dynamics of Leiden Institute of Physics, Huygens-Kamerlingh Onnes Laboratory, Leiden University (Brouwer et al, 2018; Kaczmarczyk et al, 2018). A detailed description of the apparatus is given in (Korolev et al, 2022).

The flow cell was washed with 1 ml of measurement buffer, MB (100 mM KCl, 2 mM MgCl$_2$, 10 mM NaN$_3$, 10 mM HEPES pH 7.5, 0.1% Tween 20, 0.2% BSA). The stock solutions of the #Telo-18# and 601-20 arrays (~25–30 ng/µl) labelled at the DNA ends with biotin and digoxigenin were diluted 40,000–50,000 times in the measurement buffer and attached to the streptavidin-coated 2.8 µm paramagnetic beads (Dynabeads M-280 Streptavidin, Thermo

Fisher Scientific Baltics UAB, Norway), followed by attachment to the anti-digoxygenin antibody-covered glass surface of the flow cell as described in (Korolev et al, 2022; Soman et al, 2022b). The unbound beads were removed by washing the flow cell with 1000 µl of MB at 200 µl/min. For each type of the arrays, #Telo-18# and 601-20, three series of measurements at TRF2$^{\Delta N}$ concentrations of 1, 10, and 100 nM were carried out, flushing 150 µl of the TRF2$^{\Delta N}$ dimers solution in the MB buffer into the flow cell. (The internal volume of the flow cell is 40–50 µl.) The results were compared with the data obtained in the absence of TRF2$^{\Delta N}$.

Data collection and analysis were performed using procedures and scripts in the LabVIEW environment (National Instruments, USA) described in detail in (Brouwer et al, 2021; Brouwer et al, 2020; Brouwer et al, 2018; Kaczmarczyk et al, 2018; Korolev et al, 2022; Meng et al, 2015). After manually adjusting the objective position, magnetic beads were automatically picked up and, if necessary, manually filtered, removing fault hits double and stuck beads. Typically, a trajectory of 8–0–8 mm magnet shift in 80–120 s was recorded to probe the fibres' stretch–relief response. The beads' positions in three dimensions were monitored, applying recent 2D Fast Fourier Transforms algorithms to compute cross-correlations with computer-generated reference images (Brouwer et al, 2020).

A statistical mechanics model was used to interpret the force-extension data of the nucleosome array stretching (Brouwer et al, 2018; Kaczmarczyk et al, 2018; Meng et al, 2015). The model describes the dependence of total extension of the nucleosome fibre along the direction of the applied force, $F$, as a sum of five terms attributed to different stages of the nucleosome array stretching as described in more detail in 'Results' and 'Discussion'. Experimental force–extension data were fitted to the statistical mechanics model (Brouwer et al, 2021; Kaczmarczyk et al, 2018) described in detail (Korolev et al, 2022) using both fixed and adjustable parameters. The fixed parameters are listed in Appendix Table S3.

The fitting procedure applied to all stretching curves gives the parameter values for the five adjustable parameters that are characteristic of the following properties of the fibre: (1) the total number of nucleosome particles, $N_{total}$; (2) the number of the nucleosomes not included in the stacked folded fibre, $N_{unfold}$; (3) the stiffness of folded fibre, $k_{fibre}$; (4) the free energy $\Delta G_1$ associated with the simultaneous nucleosome unstacking and partial DNA unwrapping; and 5) $\Delta G_2$, the free energy related to the second stage of DNA unwrapping up to a single turn of DNA. A detailed description is given in the references (Korolev et al, 2022; Soman et al, 2022b) and in the 'Results' section.

The number of (stacked) nucleosomes contributing to the fibre folding is defined as $N_{fold} = N_{total} - N_{unfold}$. It is not possible (Kaczmarczyk et al, 2017) to resolve the differences in the last rupture event between the nucleosomes containing complete (octamer) and incomplete (hexamer or tetramer) sets of histones, and it is assumed that both populations have the same last step in the unwrapping pathway.

The last stage of the nucleosome array stretching, the largely irreversible ruptures of 75–80 bp DNA, was analysed using a dynamic spectroscopy equation that describes the dependence of rupture force, $F_{rupt}$, on the number of nucleosomes in the array and the rate of applied force (Brower-Toland et al, 2002; Evans, 2001; Pope et al, 2005). Details are given in the 'Results' section and the Fig. EV5 legend.

## Statistical analysis

Images were analysed with FIJI/Image J (Open source, versions 2.0.0-rc-69/1.52p). Statistical analyses were performed with OriginPro (OriginLab, version 2019b). AUC data were analysed with Sedfit (Open source, version 15.01 b) and GUSSI (Open source, version 1.4.2). All experiments were independently repeated more than two times with similar results. The code used for MMT analysis is available at https://doi.org/10.5281/zenodo.6810919. Data presented are mean ± s.d. Exact $p$ values and analysis used are indicated within the corresponding figures, figure legends, and tables.

## Data availability

This study includes no data deposited in external repositories.

## Peer review information

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

## Acknowledgements

We thank the NTU Institute of Structural Biology (NISB) and Biosciences Research Centre (BRC), Nanyang Technological University, for providing access to the EM and other core facilities respectively. We are very grateful to Daniela Rhodes for the TRF2$^{\Delta N}$ construct and for the invaluable discussions. This work has been supported by the Singapore Ministry of Education (MOE) Academic Research Fund (AcRF) Tier 1 (2021-T1-002-0820) and Tier 3 (MOE2019-T3-1-012) grants.

## Author contributions

**Sook Yi Wong**: Conceptualization; Data curation; Formal analysis; Validation; Investigation; Writing—original draft; Writing—review and editing. **Aghil Soman**: Conceptualization; Formal analysis; Validation; Investigation; Writing—original draft; Writing—review and editing. **Nikolay Korolev**: Conceptualization; Formal analysis; Supervision; Validation; Investigation; Methodology; Writing—original draft; Writing—review and editing. **Wahyu Surya**: Formal analysis; Validation; Investigation. **Qinming Chen**: Investigation. **Wayne Shum**: Validation; Investigation. **John van Noort**: Conceptualization; Software; Formal analysis; Validation; Methodology; Writing—original draft; Writing—review and editing. **Lars Nordenskiöld**: Conceptualization; Supervision; Writing—original draft; Project administration; Writing—review and editing.

## Disclosure and competing interests statement

The authors declare no competing interests.

# Expanded View Figures

**Figure EV1.   TRF2$^{\Delta N}$ binds telomeric and 601 nucleosome arrays under different salt conditions.**

(**A, B**) EMSA analyses TRF2$^{\Delta N}$ binding to (**A**) Telo-10 and (**B**) 601-10 array in a solution containing 20 mM HEPES pH 7.4, 5 mM NaCl, 0.075 mM EDTA, and 0.75 mM DTT. TRF2$^{\Delta N}$ binding was done with a Telo-10 array at HO/DNA ratio of 1.05 or 601-10 array at HO/DNA ratio of 1.1 in the presence of 4% Cr$_{147}$ competitor DNA. The data shown are representatives of three technically replicated experiments. (**C, D**) EMSA analyses TRF2$^{\Delta N}$ binding to (**C**) Telo-10 and (**D**) 601-20 array in a solution containing 20 mM HEPES (pH 7.4), 62.5 mM NaCl, 0.75 mM DTT and 0.075 mM EDTA. TRF2$^{\Delta N}$ binding was done with a Telo-10 array at HO/DNA ratio of 1.05 or 601-20 DNA with 4% Cr$_{147}$ competitor DNA. The data shown are representatives of three technically replicated experiments. (**E, F**) EMSA analyses TRF2$^{\Delta N}$ binding to Telo-10 (**E**) or 601-20 (**F**) array in near-physiological conditions containing 20 mM HEPES (pH 7.4), 100 mM NaCl, 50 mM KCl, 1 mM MgCl$_2$, 0.1 mM EDTA, 1 mM DTT, 0.5 mg/ml BSA, 5% (v/v) glycerol and 0.1% (v/v) NP-40. TRF2$^{\Delta N}$ binding was done with a Telo-10 array at HO/DNA ratio of 1.05 or 601-20 DNA with 4% Cr$_{147}$ competitor DNA. The data shown are representatives of three technically replicated experiments.

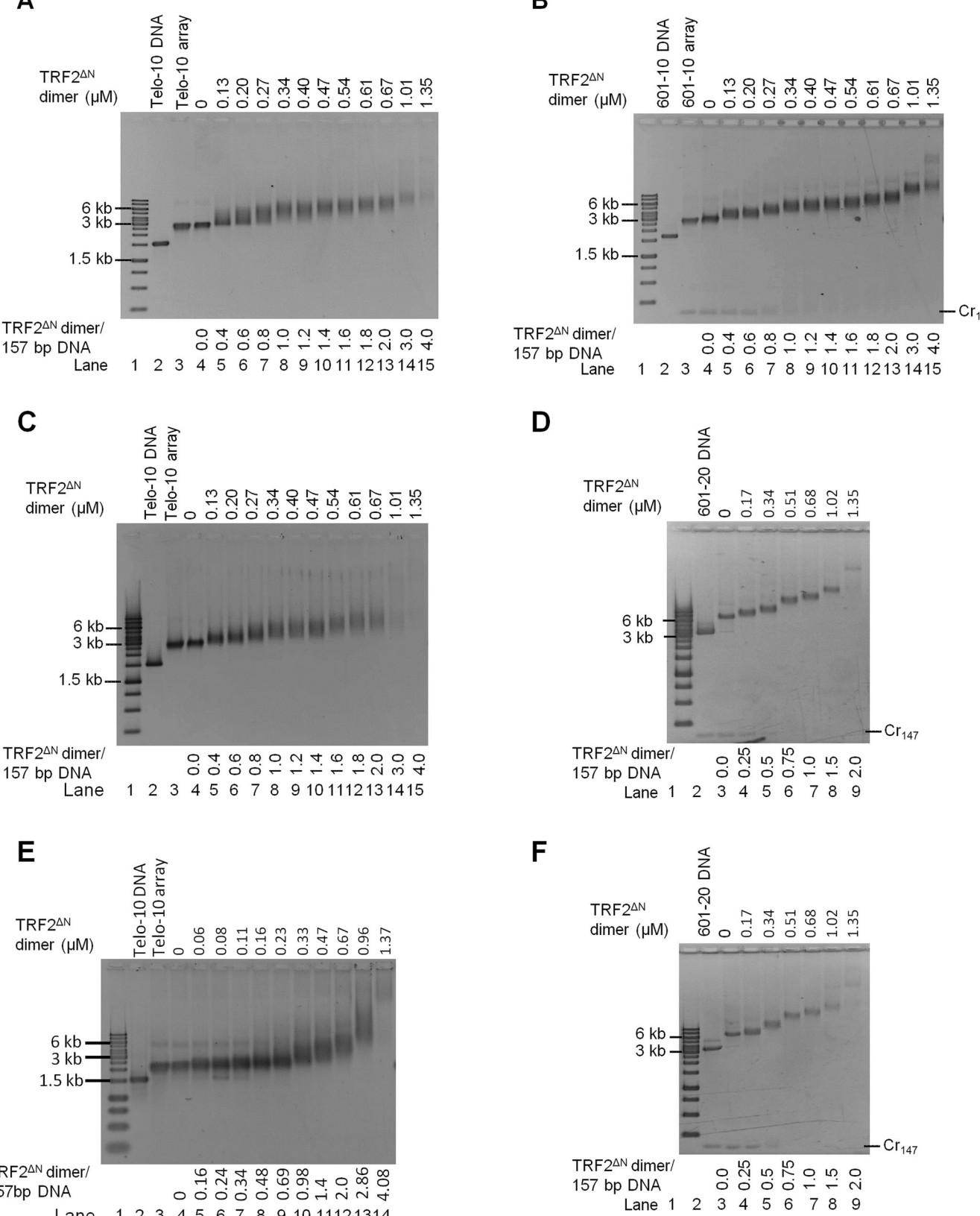

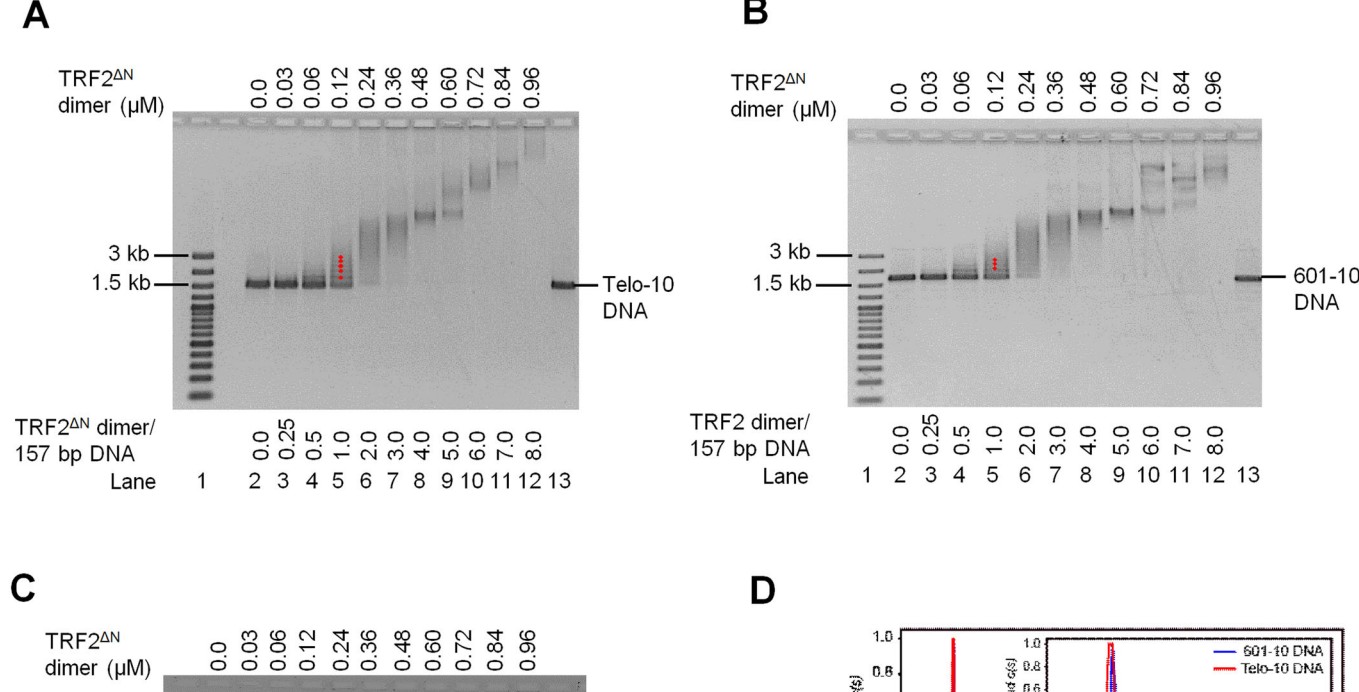

Figure EV2. **TRF2$^{\Delta N}$ binds selectively to the telomeric sequence under competing conditions.**

(A, B) EMSA analyses TRF2 binding to (A) Telo-10 DNA and (B) 601-10 DNA. The data shown are representatives of three technically replicated experiments. (C) EMSA analyses the DNA competition binding assay with TRF2$^{\Delta N}$. The amount of Telo-10 and 601-5 DNA used in the competition binding assays were at a 1:1 mass ratio. The data shown are representatives of three technically replicated experiments. (D) Sedimentation coefficient (c(s)) distribution curves, obtained from AUC-SV data, for 601-10 and Telo-10 DNA (top) and 601-10 and Telo-10 DNA in the presence of TRF2$^{\Delta N}$ (bottom). Top panel: The s-value of the Telo-10 DNA in the absence of TRF2$^{\Delta N}$ dimers was 9.49 ± 0.02 S (red, top), which shifted to 10.4 ± 0.2 S in the presence of 0.2 μM TRF2$^{\Delta N}$ dimers (corresponding to 0.6 TRF2$^{\Delta N}$ dimer/157 bp of DNA) (red, bottom). The s-value of the 601-10 DNA was 9.70 ± 0.02 S (blue, top), which shifted to 10.60 ± 0.03 S in the presence of 0.2 μM TRF2$^{\Delta N}$). Curves are the average of the three technically replicated experiments; s-values are means ± s.d.

**A**  Telo-10 array

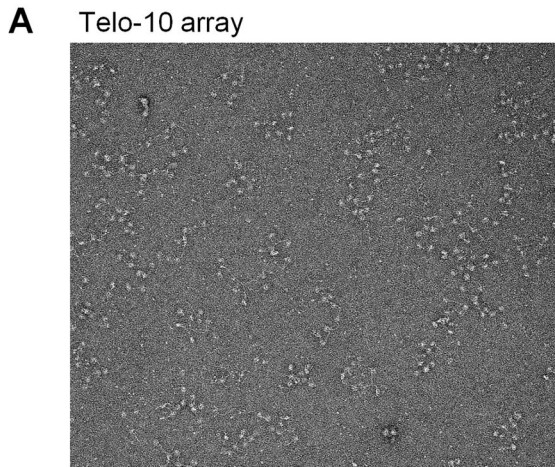

**B**  601-10 array

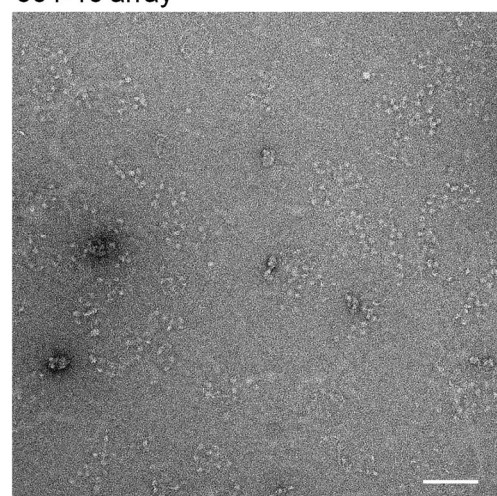

**C**

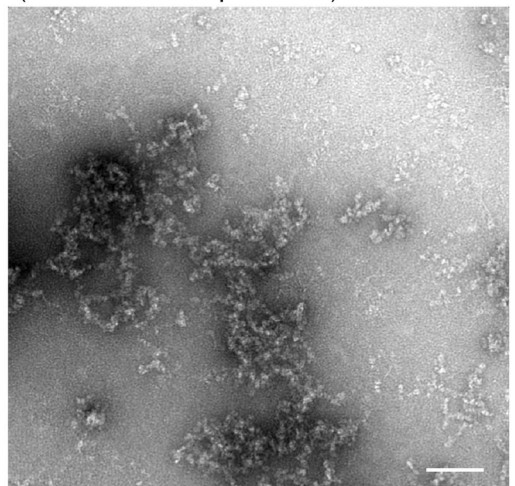

**D**  Telo-10 array + TRF2 dimer
(0.1 dimer/157 bp of DNA)

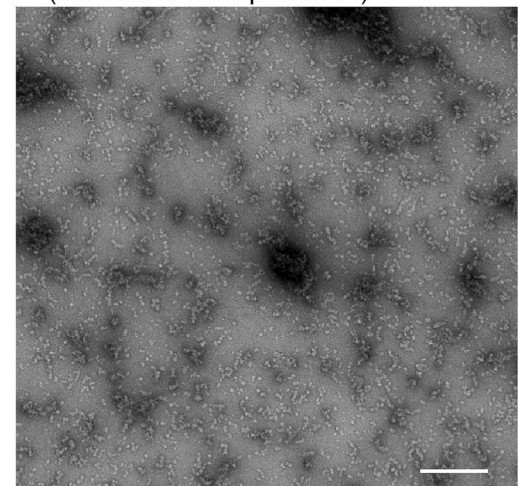

**E**  Telo-10 array + TRF2 dimer
(0.4 dimer/157 bp of DNA)

**F**  Telo-4 array + TRF2 dimer
(0.5 dimer/157 bp of DNA)

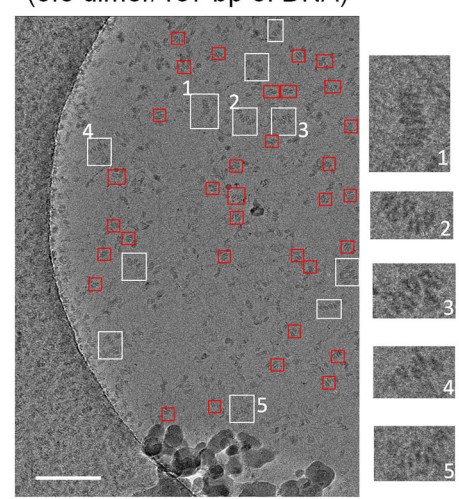

◀  **Figure EV3.   TRF2$^{\Delta N}$-telomeric array complex forms a homogeneous columnar arrangement.**

Related to Figs. 2 and 3. (**A**) Representative negative-stained EM micrographs of Telo-10 array in the absence of TRF2$^{\Delta N}$ dimer. Scale bar: 100 nm. The data shown are representatives of three technically replicated experiments. (**B**) Representative negative-stained EM micrographs of 601-10 array without TRF2$^{\Delta N}$ dimer. Scale bar: 100 nm. The data shown are representatives of three technically replicated experiments. (**C**) Histogram plot for the inter-nucleosomal distance measured from 601-10 array in the presence of TRF2$^{\Delta N}$ (blue) and Mg$^{2+}$ (purple). The insert shows the box plots overlaid with the data points (diamond symbols); statistical analysis was performed using a two-sample *t* test with an equal variance; *p* value $= 6.53 \times 10^{-9}$. Mean values are indicated in the insert; boxes show mean ± s.d.; whiskers indicate the 10-90% data range; black diamond symbols are outliers' data points. For 601-10 array in the presence of TRF2$^{\Delta N}$: $n = 413$, mean $= 15.1$ nm, s.d. $= 3.6$ nm and range $= 8.2$–$33.4$ nm. For 601-10 array in the presence of Mg$^{2+}$: $n = 419$, mean $= 13.7$ nm, s.d. $= 3.1$ nm and range $= 8.5$–$28.4$ nm. The data presented are from three technically replicated experiments. (**D**) Representative negative-stained EM micrographs of Telo-10 array in the presence of full-length TRF2 dimer at the ratio of 0.1 dimer/157 bp of telomeric DNA showing induction of columnar conformation. Scale bar: 200 nm. The data shown are representatives of three technically replicated experiments. (**E**) Representative negative-stained micrographs of Telo-10 array in the presence of full-length TRF2 dimer at the ratio of 0.4 dimer/157 bp of telomeric DNA equivalent to Fig. 2A demonstrating columnar conformation. Scale bar: 100 nm. The data shown are representatives of three technically replicated experiments. (**F**) Representative cryo-EM micrographs of Telo-4 in the presence of 0.5 TRF2 dimer/157 bp of DNA, white boxes indicate compact columnar fibres with three or more stacked nucleosomes, and red boxes indicate selected dimer stacks. Blow-ups 1–5 show columnar Telo-4 fibres in the presence of TRF2. Scale bar: 100 nm. The data shown are representatives of three technically replicated experiments.

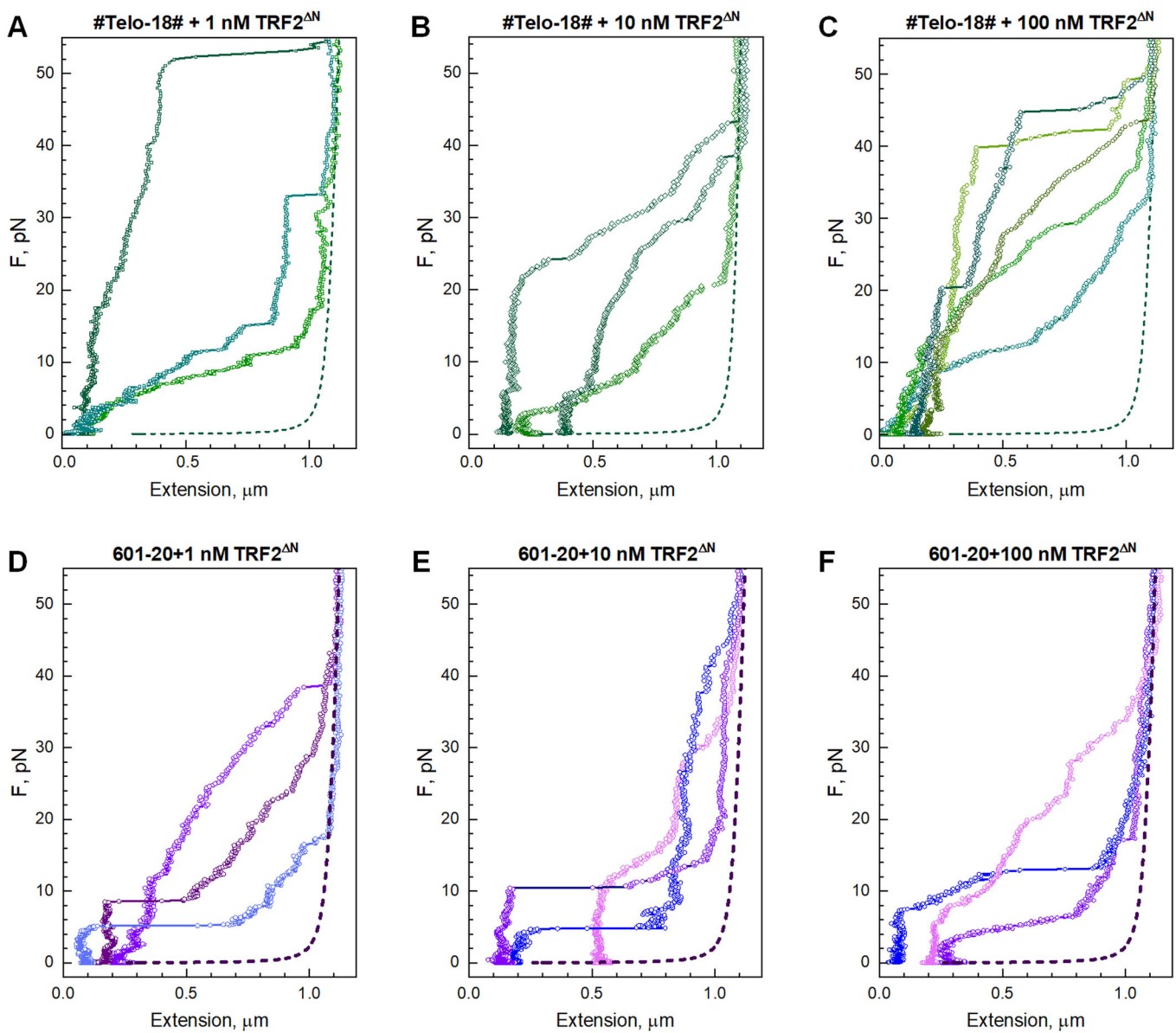

**Figure EV4. Examples of stretching curves of the #Telo-18# and 601-20 arrays which cannot be reliably fitted by the current statistical mechanics model.**

Related to Figs. 4 and 5. (A–C) Stretching curves of the #Telo-18# in the presence of 1 nM (**A**), 10 nM (**B**), and 100 nM (**C**) of the TRF2$^{\Delta N}$ dimer added to the flow cell. Dashed lines show stretching of the 3242 bp bare DNA calculated by the WLC model. (D–F) Stretching curves of the 601-20 in the presence of 1 nM (**D**), 10 nM (**E**), and 100 nM (**F**) of the TRF2$^{\Delta N}$ dimer added to the flow cell. Dashed lines show stretching of the 3242 bp bare DNA calculated by the WLC model.

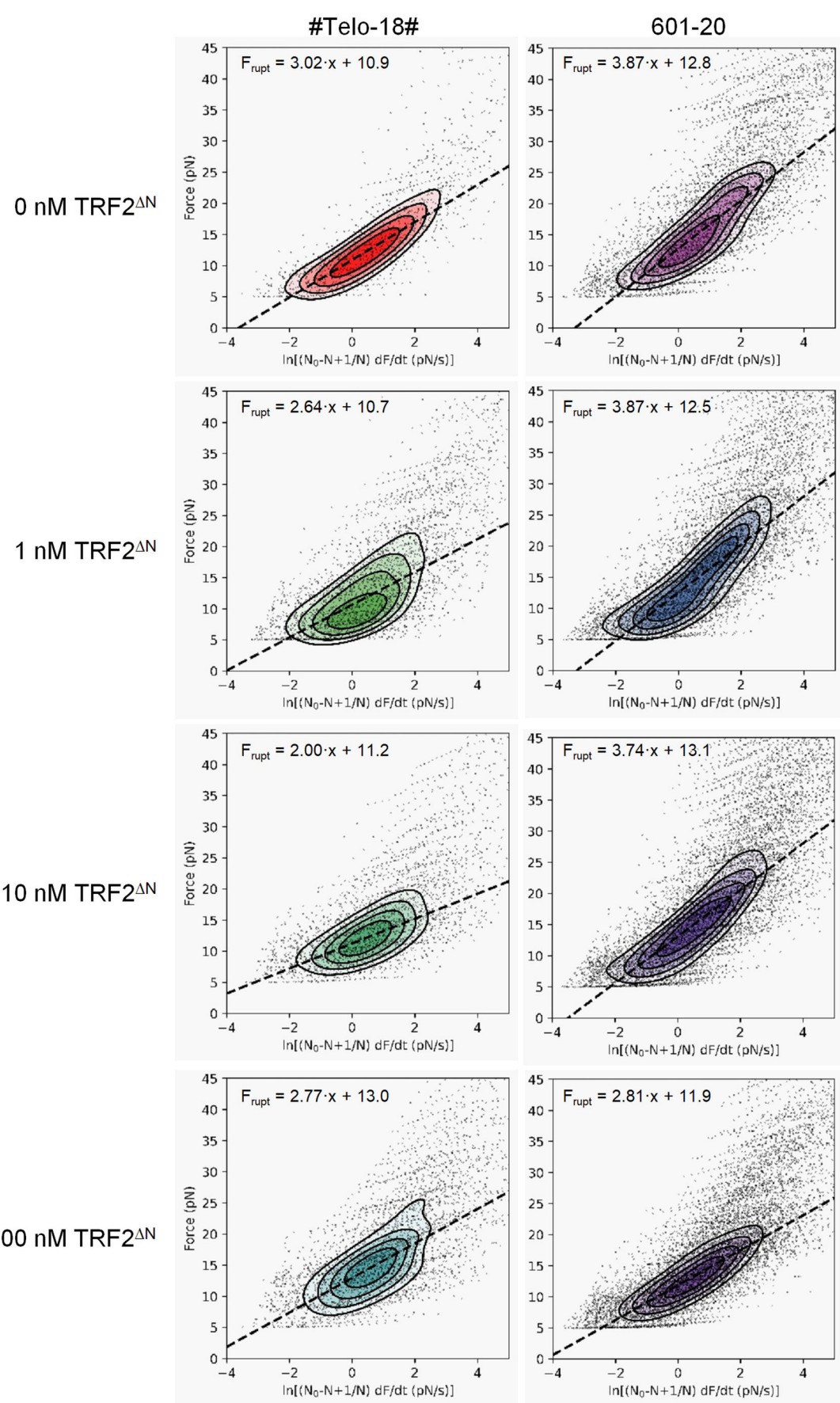

**Figure EV5.  Dependence of rupture force events, $F_{rupt}$, on the rate of applied force and the number of nucleosomes in the #Telo-18# and 601-20 arrays.**

Related to Fig. 5F. According to the equation (Pope et al, 2005): $F_{rupt} = \frac{k_B T}{d} \left\{ \ln\left[ (N_0 - N - 1/N) \frac{dF}{dt} \right] - \ln\left( \frac{k_B T k_{off}}{d} \right) \right\}$ the rupture force ($F_{rupt}$) events depend on the rate of applied force (dF/dt), the number of nucleosomes at the initial moment ($N_0$) and the moment of rupture ($N$). In the equation above, $d$ is the distance between the bound state and the activation barrier peak along the direction of the applied force; $k_{off}$ is the rate constant for nucleosome rupture under zero external force; $k_B$ is the Boltzmann constant, $T$ is temperature. Note, that the pre-factor was added in the logarithm, as compared to (Pope et al, 2005), to adjust for force clamp mode as opposed to optical tweezers that operate in position clamp mode. Values of the $d$ and $k_{off}$, which are displayed in Fig. 5F of the main text, were determined from the slope and intercept of the linear fit in coordinates $F_{rupt}$ versus $x = \ln\left[ (N_0 - N - 1/N) \frac{dF}{dt} \right]$. In each graph, points indicate rupture events, and coloured contours highlight densities of the rupture distribution (2D histograms). For a more representative linear fit, the large impact of outliers was reduced by including only datapoints in bins that exceed a threshold of 0.5 times the peak value in the 2D histogram. The coloured contours indicate the areas of points included in the linear fitting; this 0.5-maximum cut-off discards about 50% of the datapoints. Lines show the linear fit of the data. The left-hand column of four graphs presents the data collected for the #Telo-18# arrays as indicated at the top; four graphs on the right are the results for the 601-20 arrays. Concentrations of the TRF2$^{\Delta N}$ dimer in the flow cell are 0, 1, 10, and 100 nM, as indicated in the four rows of two graphs. Numerical values (including mean ± s.d. and numbers of measured points, $n$) are given in Appendix Table S2.

