## [Peer Review File · The EMBO Journal]

The shelterin component TRF2 mediates columnar stacking of human telomeric chromatin

Sook Yi Wong, Aghil Soman, Nikolay Korolev, Wahyu Surya, Qinming Chen, Wayne Shum, John Van Noort, and Lars Nordenskiöld

DOI: [10.15252/emboj.2023114491](https://doi.org/10.15252/emboj.2023114491)

Corresponding author: Lars Nordenskiöld (LarsNor@ntu.edu.sg)

Review Timeline:

Submission Date:	14th May 23
Editorial Decision:	23rd Jun 23
Revision Received:	20th Sep 23
Editorial Decision:	13th Oct 23
Revision Received:	24th Oct 23
Accepted:	26th Oct 23

Editor: Daniel Klimmeck

Transaction Report:

Dear Dr Nordenskiöld,

Thank you again for the submission of your manuscript (EMBOJ-2023-114491) to The EMBO Journal, as well as for your patience with our response at this time of the year which got protracted due to delayed referee input and discussions in the editorial team. Your study has been sent to two reviewers with expertise in telomere and chromatin structural biology and we have received feedback from both of them, which I enclose below.

As you will see, the referees acknowledge the potential interest and novelty of your comparative analysis, although they also express several issues that will have to be conclusively addressed before they can be supportive of publication of your manuscript in The EMBO Journal. We judge the comments of the referees to be generally reasonable and given their overall interest, we are happy to invite you to revise your manuscript experimentally to address the referees' comments.

As you may have seen on our web page, we generally allow three months as standard revision time. As a matter of policy, competing manuscripts published during this period will not negatively impact on our assessment of the conceptual advance presented by your study. However, we request that you contact the editor as soon as possible upon publication of any related work, to discuss how to proceed. Should you foresee a problem in meeting this three-month deadline, please let us know in advance and we may be able to grant an extension.

When submitting your revised manuscript, please carefully review the instructions below.

Please feel free to approach me any time should you have any questions related to this.

Thank you for the opportunity to consider your work for publication.

I look forward to your revision.

Kind regards,

Daniel Klimmeck

Daniel Klimmeck, PhD
Senior Editor
The EMBO Journal

Instruction for the preparation of your revised manuscript:

- 1) a .docx formatted version of the manuscript text (including legends for main figures, EV figures and tables). Please make sure that the changes are highlighted to be clearly visible.
- 2) individual production quality figure files as .eps, .tif, .jpg (one file per figure).
- 3) a .docx formatted letter INCLUDING the reviewers' reports and your detailed point-by-point response to their comments. As part of the EMBO Press transparent editorial process, the point-by-point response is part of the Review Process File (RPF), which will be published alongside your paper.
- 4) a complete author checklist, which you can download from our author guidelines ([https://wol-prod-cdn.literatumonline.com/pb-assets/embo-site/Author Checklist%20-%20EMBO%20J-1561436015657.xlsx](https://wol-prod-cdn.literatumonline.com/pb-assets/embo-site/Author%20Checklist%20-%20EMBO%20J-1561436015657.xlsx)). Please insert information in the checklist that is also reflected in the manuscript. The completed author checklist will also be part of the RPF.
- 5) Please note that all corresponding authors are required to supply an ORCID ID for their name upon submission of a revised manuscript.
- 6) It is mandatory to include a 'Data Availability' section after the Materials and Methods. Before submitting your revision, primary datasets produced in this study need to be deposited in an appropriate public database, and the accession numbers and database listed under 'Data Availability'. Please remember to provide a reviewer password if the datasets are not yet public (see <https://www.embopress.org/page/journal/14602075/authorguide#datadeposition>). In case you have no data that requires deposition in a public database, please state so in this section. Note that the Data Availability Section is restricted to new primary data that are part of this study.

7) Our journal encourages inclusion of *data citations in the reference list* to directly cite datasets that were re-used and obtained from public databases. Data citations in the article text are distinct from normal bibliographical citations and should directly link to the database records from which the data can be accessed. In the main text, data citations are formatted as follows: "Data ref: Smith et al, 2001" or "Data ref: NCBI Sequence Read Archive PRJNA342805, 2017". In the Reference list, data citations must be labeled with "[DATASET]". A data reference must provide the database name, accession number/identifiers and a resolvable link to the landing page from which the data can be accessed at the end of the reference. Further instructions are available at .

8) At EMBO Press we ask authors to provide source data for the main and EV figures. Our source data coordinator will contact you to discuss which figure panels we would need source data for and will also provide you with helpful tips on how to upload and organize the files.

Numerical data can be provided as individual .xls or .csv files (including a tab describing the data). For 'blots' or microscopy, uncropped images should be submitted (using a zip archive or a single pdf per main figure if multiple images need to be supplied for one panel). Additional information on source data and instruction on how to label the files are available at .

9) We replaced Supplementary Information with Expanded View (EV) Figures and Tables that are collapsible/expandable online (see examples in <https://www.embopress.org/doi/10.15252/embj.201695874>). A maximum of 5 EV Figures can be typeset. EV Figures should be cited as 'Figure EV1, Figure EV2' etc. in the text and their respective legends should be included in the main text after the legends of regular figures.

10) When assembling figures, please refer to our figure preparation guideline in order to ensure proper formatting and readability in print as well as on screen:
<http://bit.ly/EMBOPressFigurePreparationGuideline>

11) For data quantification: please specify the name of the statistical test used to generate error bars and P values, the number (n) of independent experiments (specify technical or biological replicates) underlying each data point and the test used to calculate p-values in each figure legend. The figure legends should contain a basic description of n, P and the test applied. Graphs must include a description of the bars and the error bars (s.d., s.e.m.).

We realize that it is difficult to revise to a specific deadline. In the interest of protecting the conceptual advance provided by the work, we recommend a revision within 3 months (21st Sep 2023). Please discuss the revision progress ahead of this time with the editor if you require more time to complete the revisions.

Referee #1:

The recent identification of the columnar chromatin structure on telomeric DNA (by the same authors) was a breakthrough, because of its unique and unexpected architecture. This finding left several new open questions. First and far most, how this structure forms at telomeres, where several proteins are bound?

This manuscript by Wong, Soman et al. studies the effect of the TRF2 protein on telomeric nucleosomal arrays. The authors find that the TRF2 Δ N construct can induce formation of the columnar telomeric chromatin structure, suggesting that binding of TRF2 may induce the formation of this structure. The authors use a combination of biochemical, biophysical and structural techniques to test their hypotheses. Generally, the data is of high quality and it supports the conclusions. I however have few comments.

1. Can the authors include a DNA only control in their AUC experiment? It is not clear if the effects seen are due simply to nucleosomes, or if the DNA binding properties of TRF2 Δ N affect the results. Also, The EM data shows that TRF2 Δ N induces different conformations on the Telo or 601 arrays, however these arrays behave similarly in AUC when TRF2 Δ N is added, can the authors clarify this?
2. Figure 5 is difficult to grasp for non-experts. I am not sure I see the effects that are suggested by the authors, as it seems simply that the green data points spread wider than the red ones, but I am not sure how specific this is for high forces only. Also, this experiment should include a control with the 601 chromatin array to prove the specificity of the observed effect for the telomeric array. A control with 601-arrays is important also for Figure 4.
3. In general, the statement that TRF2 Δ N binds the supergroove is purely speculative based on the data. The authors mention this throughout the manuscript as a tentative proposition, and they include a statement in the abstract. Based on the complete lack of evidence for this specific binding mode, and the possibility that other TRF2 binding modes induce this chromatin structure, I suggest that statement to be taken out of the abstract.

Referee #2:

In this manuscript, Wong et al. address an unresolved issue in telomere biology, namely how shelterin - the protein complex that protects telomeres - interplays with telomeric chromatin. The authors use analytical ultracentrifugation sedimentation velocity (AUC-SV), electron microscopy (EM), and single molecule force spectroscopy to study the in vitro interaction of one of the components of shelterin, Telomere repeat binding factor 2 (TRF2), with telomeric nucleosomal arrays. They present convincing evidence that TRF2 induces the compaction and stabilization of telomeric chromatin fibers. These observations have interesting implications for telomere structure and function and give an important contribution to the field. However, some major and minor issues should be addressed.

1. Figure S2 shows that TRF2 Δ N binds efficiently not only Telo-10 nucleosomal array but also 601-10 and 601-20 arrays. As the authors point out, this is an unexpected result. As a possible explanation, the authors cite the results by Mukherjee et al. showing that TRF2 binds in several extratelomeric sites on the genome. However, Mukherjee et al. report that TRF2 binds essentially G-quadruplex forming sequences, and in the 601 DNA sequence there are not any apparent telomeric repeats nor G4 forming sequences. The TRF2 Δ N lacks most of the basic N-terminal region of TRF2 that might give rise to unspecific binding (excluded by the authors by testing different salt conditions (Fig. S2)). However, it seems to me that non-specific binding cannot be excluded. Even if TRF2 Δ N binds to 601 DNA sequence, I expect it does with a lower affinity with respect to telomeric DNA. The EMSA assays reported in Fig. S2 do not show any apparent difference between Telo-10 and 601-10 arrays. Is it possible that TRF2 Δ N recognizes structural features of nucleosomal arrays? It would be helpful that the authors show EMSA analyses of the binding of TRF2 Δ N to Telo-10 and 601-10 naked DNA (or to 157bp Telo and 601 monomers).
2. Authors show that TRF2 full-length induces the columnar structure of telomeric chromatin similarly to TRF2 Δ N. Did the authors try TRF2 full-length also with 601-10 arrays?
3. In the legends of Fig. 2 and Fig. S4, the authors should report whether it is EM or Cryo-EM micrographs for each figure.

Response to referees' comments

on the manuscript "*The shelterin component TRF2 mediates the columnar stacking of human telomeric chromatin*" by S. Y. Wong et al. submitted to EMBO Journal (MS ID: **EMBOJ-2023-114491**)

We are very grateful to the reviewers for the valuable criticism and suggested corrections and for suggesting additional controls to strengthen our case. To address the referees' concerns, we conducted several additional experiments and answered all queries:

To clarify concern 1.1 (reviewer 1), we included a DNA only control in the AUC experiments.

To address comment 1.2, we included a control with the 601-chromatin array to prove the specificity of the observed effect for the telomeric array.

To comply with concern 2.1 (reviewer 2), we performed EMSA analyses of the binding of TRF2^{ΔN} to DNA with telomeric (Telo-10) and nucleosome positioning (601-10) sequences.

The additional measurements confirm and strengthen the results and conclusions in the submitted work. A detailed description of the additional experiments is given below in response to the reviewers' comments. We believe that the changes we have now made have significantly improved this manuscript. We rest assured that the manuscript is now suitable for publication in the EMBO Journal.

Below, we answer each point raised by the reviewers. **Our responses are highlighted in blue font. In the revised manuscript, changes are marked in red font.** Furthermore, we performed other minor stylistic and linguistic changes.

Referee #1:

"The recent identification of the columnar chromatin structure on telomeric DNA (by the same authors) was a breakthrough, because of its unique and unexpected architecture. This finding left several new open questions. First and far most, how this structure forms at telomeres, where several proteins are bound?"

This manuscript by Wong, Soman et al. studies the effect of the TRF2 protein on telomeric nucleosomal arrays. The authors find that the TRF2dN construct can induce formation of the columnar telomeric chromatin structure, suggesting that binding of TRF2 may induce the formation of this structure. The authors use a combination of biochemical, biophysical and structural techniques to test their hypotheses. Generally, the data is of high quality and it supports the conclusions. I however have few comments. "

1.1. "Can the authors include a DNA only control in their AUC experiment? It is not clear if the effects seen are due simply to nucleosomes, or if the DNA binding properties of TRF2dN affect the results. Also, The EM data shows that TRF2dN induces different conformations on the Telo or 601 arrays, however these arrays behave similarly in AUC when TRF2dN is added, can the authors clarify this?"

We are grateful for this suggestion of the referee and carried out additional AUC-SV measurements comparing the sedimentation behaviour of the Telo-10 DNA and 601-10 DNA in the absence and the presence of the TRF2^{ΔN} dimer (0.2 μM TRF2^{ΔN} corresponding to 0.6 TRF2^{ΔN}

dimer/157 bp DNA). This new AUC-SV data has been included in the revised manuscript (Extended View Fig. EV2, panel D) and discussed in the revised manuscript (p. 10). In the presence of TRF2^{ΔN}, only a slight increase in the s-value was observed for both Telo-10 and 601-10 DNAs. Notably, the addition of TRF2^{ΔN} resulted in broadening the $s_{20,w}$ Telo-10 and 601-10 distributions, which implied heterogeneous populations of the TRF2^{ΔN}-DNA complex. This result suggests that the narrowing of the $s_{20,w}$ profile upon TRF2^{ΔN} addition that we observed for the Telo-10 chromatin only occurs with telomeric nucleosomes. Interaction of the TRF2^{ΔN} dimer with telomeric DNA alone is insufficient to promote the narrowing of the $s_{20,w}$ distribution, suggesting a more homogeneous fibre structure observed for the Telo-10 array in the presence of TRF2^{ΔN}.

To clarify the second part of the referee's question regarding "*The EM data shows that TRF2dN induces different conformations on the Telo or 601 arrays; however these arrays behave similarly in AUC when TRF2dN is added*", we emphasised more clearly that in fact, the narrowing of the s-value distribution was observed only when TRF2^{ΔN} was added to the Telo-10 array and not with the 601-10 array (p. 9). This narrowing indicates that the Telo-10 array was more homogeneous in the presence of TRF2^{ΔN}, and it is in line with the EM observations, in which we observe that in the presence of TRF2^{ΔN}, a large proportion of the Telo-10 fibre (82.5%) appears as regular columnar structures. On the other hand, the AUC-SV s-value profiles of the 601-10 arrays exhibit a wider distribution upon the addition of TRF2^{ΔN}, indicating a heterogeneous fibre population. This AUC-SV result is in agreement with our EM data that shows variable structures of the TRF2^{ΔN}-601-10 nucleosome array complex: 31% globular and 69% ladder-like, with ladder structures showing different degrees of compactness.

1.2. "Figure 5 is difficult to grasp for non-experts. I am not sure I see the effects that are suggested by the authors, as it seems simply that the green data points spread wider than the red ones, but I am not sure how specific this is for high forces only. Also, this experiment should include a control with the 601 chromatin array to prove the specificity of the observed effect for the telomeric array. A control with 601-arrays is important also for Figure 4."

We are thankful for the referee's suggestion to include a control with 601 arrays. We carried out a series of additional single molecule force spectroscopy MMT measurements with 601-20 arrays in the presence and absence of the TRF2^{ΔN} protein. The experimental conditions were similar to those with the #Telo-18# arrays. This new data is added to the revised manuscript. Furthermore, the figures presenting the MMT data were rearranged to accommodate the 601-20 arrays data. Examples of 601-20 array stretch-relief curves were added to Fig. 4; the summary of the 5 parameters (N_{total} , N_{fold} , k_{fibre} , ΔG_1 , and ΔG_2) of the #Telo-18# arrays were shifted to Fig. 5 and combined with similar data for the 601-20 arrays (panels A-E of revised Fig. 5). Results of the 601-20 arrays study were added to the Appendix Tables S2 and S3.

The major finding of the 601-20 array MMT study is that the addition of the TRF2^{ΔN} dimer does not have a major influence on the mechanical properties of these arrays, even at 100 nM protein concentration in the flow cell. This is in clear and remarkable contrast to the #Telo-18# array data, which shows a noticeable increase in resistance to the applied stretching force already in the presence of 1 nM TRF2^{ΔN} dimer. However, we suggest that the TRF2^{ΔN} protein binds to the 601-20 arrays in a non-specific manner driven by primarily electrostatic interactions. This binding mode produces a slight increase of the N_{total} , N_{fold} , ΔG_1 , and k_{fibre} values observed at 100 nM TRF2^{ΔN} dimer (Fig. 5), as well as an appearance of a small fraction of the data with higher forces at the initial stage of fibre stretching that cannot be fitted to the statistical mechanics model (Fig. EV4).

Furthermore, we performed an additional in-depth analysis of the rupture force (F_{rupt}) data collected for the #Telo-18# and 601-20 arrays in the presence and the absence of the TRF2^{ΔN} dimer. Rupture is an abrupt unwinding of the inner DNA turn in the nucleosome observed at high applied force. The scatter plots of F_{rupt} versus $x = \ln[(N_0 - N + 1/N) \cdot dF/dt]$ and their linear fittings for both the

#Telo-18# and the new 601-20 array data were drawn as a new figure and presented as Extended View Fig. EV5. To address the reviewer's concern about the difficulty of grasping the graphs for the respective $F_{rupt} - x$ scatter plots, they were combined with contour graphs, highlighting the highest density of points. Also, we avoid displaying the overlapping of data obtained in the presence and absence of the TRF2^{ΔN}. For a more robust fit, the impact of outliers were reduced by including only data points in bins that exceeded a threshold in the 2D histogram of 0.5 times the peak value (indicated as the coloured contours in Fig EV5). Application of this cut-off discards about 50% of the datapoints. Comparison with previously published data sets for the #Telo-18# and 601-20 arrays without TRF2^{ΔN} (*Nature*, 2022, 609:1048) yielded only a small difference in the rupture characteristics.

The parameters k_{off} and d for the #Telo-18# and 601-20 arrays (calculated from the linear fittings of the F_{rupt} versus x) are presented as panel F of Fig. 5. For the #Telo-18# arrays, presence of TRF2^{ΔN} increases the rupture length d from about 1.1 to up to 2.1 nm (at 10 nM TRF2^{ΔN}) and decreases the rupture rate k_{off} , from about $8.9 \cdot 10^{-3}$ down to $1.8 \cdot 10^{-3} \text{ s}^{-1}$ (at 10 nM TRF2^{ΔN}). The rupture length decreased at 100 nM TRF2^{ΔN}, possibly by the effect of multiple TRF2^{ΔN} protein binding to the telomeric nucleosome. Nucleosomes in the 601-20 array were less affected by the TRF2^{ΔN} addition. Only at 100 nM TRF2^{ΔN}, did both the rupture length and the rupture rate of the 601 nucleosomes change and approach the level of the telomeric nucleosomes (see Fig 5F and Appendix Table S3). Overall, the analysis of the rupture forces showed that TRF2^{ΔN} has a stabilising effect on nucleosomes, which is significantly more pronounced on the telomeric than on the 601 nucleosomes. Our d and k_{off} values are in the range of the earlier published results determined under significantly different settings (different solution conditions, dissimilar NRL of the arrays, optical versus magnetic tweezers) (*Nucleic Acids Res.*, 2018, 47:666; *Cell Rep.*, 2022, 41:111858). In these studies, the addition of the nucleosome-destabilising proteins (HMGB or FACT) leads to the expected increase of the k_{off} value (named opening rate in the above-cited papers), while in our work, stabilising influence of the TRF2^{ΔN} protein is reflected by the k_{off} decrease.

The changes addressing the referee's comment, presentation and discussion of the additional data are given on p.16-18 of the Results section, p. 25 of the Methods section and in Fig 5F and Fig 5EV and their legends.

1.3. "In general, the statement that TRF2dN binds the supergroove is purely speculative based on the data. The authors mention this throughout the manuscript as a tentative proposition, and they include a statement in the abstract. Based on the complete lack of evidence for this specific binding mode, and the possibility that other TRF2 binding modes induce this chromatin structure, I suggest that statement to be taken out of the abstract."

We agree with the referee's note and removed the statement from the abstract. We also changed some formulations in the main text to make clear that the suggested binding mode of the TRF2^{ΔN} protein is putative.

Referee #2

"In this manuscript, Wong et al. address an unresolved issue in telomere biology, namely how shelterin - the protein complex that protects telomeres - interplays with telomeric chromatin. The authors use analytical ultracentrifugation sedimentation velocity (AUC-SV), electron microscopy (EM), and single molecule force spectroscopy to study the in vitro interaction of one of the components of shelterin, Telomere repeat binding factor 2 (TRF2), with telomeric nucleosomal arrays. They present convincing evidence that TRF2 induces the compaction and stabilisation of

telomeric chromatin fibers. These observations have interesting implications for telomere structure and function and give an important contribution to the field. However, some major and minor issues should be addressed."

2.1. "Figure S2 shows that TRF2 Δ N binds efficiently not only Telo-10 nucleosomal array but also 601-10 and 601-20 arrays. As the authors point out, this is an unexpected result. As a possible explanation, the authors cite the results by Mukherjee et al. showing that TRF2 binds in several extratelomeric sites on the genome. However, Mukherjee et al. report that TRF2 binds essentially G-quadruplex forming sequences, and in the 601 DNA sequence there are not any apparent telomeric repeats nor G4 forming sequences. The TRF2 Δ N lacks most of the basic N-terminal region of TRF2 that might give rise to unspecific binding (excluded by the authors by testing different salt conditions (Fig. S2)). However, it seems to me that non-specific binding cannot be excluded. Even if TRF2 Δ N binds to 601 DNA sequence, I expect it does with a lower affinity with respect to telomeric DNA. The EMSA assays reported in Fig. S2 do not show any apparent difference between Telo-10 and 601-10 arrays. Is it possible that TRF2 Δ N recognises structural features of nucleosomal arrays? It would be helpful that the authors show EMSA analyses of the binding of TRF2 Δ N to Telo-10 and 601-10 naked DNA (or to 157bp Telo and 601 monomers)."

We are very thankful to the referee for this thorough and well-substantiated comment. In response to her/his note as well as the other reviewer's concerns, we carried out additional experiments to compare TRF2 Δ N binding to telomeric and non-telomeric ("601") DNA. The EMSA analyses of TRF2 Δ N binding to Telo-10 and 601-10 naked DNA showed a slight preference for telomeric DNA. Nonetheless, under competing conditions, TRF2 Δ N had a clear binding preference for the telomeric DNA, indicating its sequence-specific preference. Regardless, the columnar arrangement observed with the Telo-10 array in the presence of full-length TRF2 or TRF2 Δ N was unique and unambiguous to telomeric nucleosomes as the additional AUC experiments on TRF2 Δ N binding to Telo-10 and 601-10 DNAs did not report homogeneous compaction of the DNAs. Please see the detailed response to the reviewer 1. These new results are described in the revised manuscript (p. 7 and 10). As highlighted by the referee, the paper by Mukherjee et al. (*J. Biol. Chem.*, 2019, 294:17709) refers to the G-quadruplex-forming sequences. However, this study also stated that it was not possible to identify a consensus sequence motif for the TRF2 binding. In addition, an earlier study (*Cell Res.*, 2011, 21:1028) reports that TRF1 and TRF2 bind to non-telomeric DNA sequences. Mukherjee's study further implies that "... our results suggest that TRF2 binds to DNA structure instead of a sequence motif". We speculate that the highly bendable 601 DNA with periodic TA elements form 'DNA structures' that might be bound to TRF2. We agree that there might be some non-specific interactions involved, as Rap1, the binding partner of TRF2 *in vivo*, has been shown to increase the specificity of TRF2 to telomeric sequences by suppressing non-specific interactions (*Nucleic Acids Res.*, 2015, 43:2691). Our system lacks Rap1, which might be the reason for the observed non-specific interaction. We have made the reference to the Mukherjee study more evident and articulated the above in the main text. (p. 7).

2.2. "Authors show that TRF2 full-length induces the columnar structure of telomeric chromatin similarly to TRF2 Δ N. Did the authors try TRF2 full-length also with 601-10 arrays?"

We did not perform experiments on full-length TRF2 with 601-10 arrays. Given that the EM results for the Telo-10 arrays complexed with TRF2 Δ N and full-length TRF2 were largely similar, it gave us reasonable grounds to assume that there would not be a significant difference when using full-length TRF2 with a 601-10 array. Although the lack of the basic N-terminal in the TRF2 Δ N could alter the affinity of the protein to the nucleosome, the binding of the TRF2 Δ N to nucleosomes is still primarily mediated by the myb domain that is unchanged in the TRF2 Δ N. As evident from the DNA competition binding assay (Figure EV2C), TRF2 Δ N retains its high affinity for the telomeric sequences. Moreover, the basic domain of the TRF2 is mainly involved in regulating other cellular processes,

particularly in the DNA repair pathways, rather than being functionally involved in the binding to the 'bulk' DNA or nucleosomes. Hence, we believe that binding full-length TRF2 to the 601-10 arrays would not significantly differ from the one we observed with TRF2^{ΔN}. Our main finding is that TRF2^{ΔN} and full-length TRF2 compact and stabilise the columnar structure of the telomeric chromatin. This was not observed for the 601 arrays in the presence of TRF2^{ΔN} that instead induced the formation of the ladder-like zig-zag structures. It is an expected result since the 601 DNA is a positioning sequence where the nucleosomes are forced to be separated by a linker DNA of designed length to define the nucleosome repeat length, NRL, precisely. In our study, NRL = 157 bp, the linker DNA is 10 bp. This prohibits the formation of the closely stacked column of nucleosomes; these columns can form only if the effective NRL is about 132 bp. Hence, neither TRF2^{ΔN} nor full-length TRF2^{ΔN} would have the capacity to induce or stabilise the columnar form, and our study has indeed shown that for the case of TRF2^{ΔN}.

2.3. "In the legends of Fig. 2 and Fig. S4, the authors should report whether it is EM or Cryo-EM micrographs for each figure."

We made the suggested changes in the legends of Fig 2 and Figure EV3.

Dear Dr Nordenskiöld,

Thank you for submitting your revised manuscript (EMBOJ-2023-114491R1) to The EMBO Journal, as well as for your patience with our response at this time of the year. Your amended study was sent back to the two referees for their re-evaluation, and we have received comments from both of them, which I enclose below.

As you will see, the experts stated that the work has been substantially improved by the revisions and they are now in broadly favour of publication, pending minor revision.

Thus, we are pleased to inform you that your manuscript has been accepted in principle for publication in The EMBO Journal.

Please consider the remaining minor requests by referee #2 carefully and adjust the manuscript accordingly where appropriate.

Also, we now need you to take care of a number of issues related to formatting and data presentation as detailed below, which should be addressed at re-submission.

Please contact me at any time if you have additional questions related to below points.

As you might have seen on our web page, every paper at the EMBO Journal now includes a 'Synopsis', displayed on the html and freely accessible to all readers. The synopsis includes a 'model' figure as well as 2-5 one-short-sentence bullet points that summarize the article. I would appreciate if you could provide this figure and the bullet points.

Thank you for giving us the chance to consider your manuscript for The EMBO Journal. I look forward to your final revision.

Again, please contact me at any time if you need any help or have further questions.

Kind regards,

Daniel Klimmeck

>> Please adjust the title of the 'Competing Interests' section to 'Disclosure and Competing Interests Statement'.

>> Figure callouts: are currently in the wrong order for Fig 2B (called out after Fig 3A); Fig 4 F-H (called out after Fig 5); Appendix Table S1 (called out after Table S3) and need to be corrected.

>> Data accessibility section: please deposit the newly acquired Cryo-EM density maps in the EMD database and make them publically available, providing html web link and data set ID. Adjust the author checklist accordingly.

>> Appendix file: amend the ToC on the first page with page numbers.

>> Please enter a separate 'Statistical Analysis' section in the 'Material and Methods' part, detailing all algorithms and significance measures applied.

>> Consider additional changes and comments from our production team as indicated below:

- Figure legends:

Please note that the box plots need to be defined in terms of minima, maxima, centre, bounds of box and whiskers, and percentile in the legend of figures 3d.

Please note that information related to n is missing in the legend of figures 3c-d; 5a-f; EV3a

Referee #1:

The authors have addressed my comments with new experiments and clarifications. This has improved the manuscript which is now ready for publication. Congratulations on an exciting story.

Referee #2:

In their revision, the authors have addressed the major points raised by both referees. I think the manuscript is substantially improved and is now suitable for publication in EMBO Journal.

Minor points:

Page 7, line 3: the authors cited the wrong figure (I guess it is Fig EV1B/D/F instead of Fig EV2B/D/E)

Page 13, line 23: preparation of full-length TRF2 is shown in Fig S1F, instead of Fig S2F.

Referee #1:

The authors have addressed my comments with new experiments and clarifications. This has improved the manuscript which is now ready for publication. Congratulations on an exciting story.

We are grateful to Reviewer 1 for the positive assessment.

Referee #2:

In their revision, the authors have addressed the major points raised by both referees. I think the manuscript is substantially improved and is now suitable for publication in EMBO Journal.

We are grateful to Reviewer 2 for the positive assessment.

Minor points:

Page 7, line 3: the authors cited the wrong figure (I guess it is Fig EV1B/D/F instead of Fig EV2B/D/E)

Page 13, line 23: preparation of full-length TRF2 is shown in Fig S1F, instead of Fig S2F.

This is rectified.

Dear Dr Nordenskiöld,

Thank you for submitting the revised version of your manuscript. I have now evaluated your amended manuscript and concluded that the remaining minor concerns have been sufficiently addressed.

Thus, I am pleased to inform you that your manuscript has been accepted for publication in the EMBO Journal.

Please note that it is EMBO Journal policy for the transcript of the editorial process (containing referee reports and your response letter) to be published as an online supplement to each paper.

Also, in case you might NOT want the transparent process file published at all, you will also need to inform us via email immediately. More information is available here:

<https://www.embopress.org/page/journal/14602075/authorguide#transparentprocess>

Please note that in order to be able to start the production process, our publisher will need and contact you shortly regarding the page charge authorisation and licence to publish forms.

Authors of accepted peer-reviewed original research articles may choose to pay a fee in order for their published article to be made freely accessible to all online immediately upon publication. The EMBO Open fee is fixed at \$6,540 USD / £5,310 GBP / €5,900 EUR (+ VAT where applicable), pending application of a waiver which might be applicable in this case as discussed.

Should you be planning a Press Release on your article, please get in contact with embojournal@wiley.com as early as possible, in order to coordinate publication and release dates.

On a different note, I would like to alert you that EMBO Press is currently developing a new format for a video-synopsis of work published with us, which essentially is a short, author-generated film explaining the core findings in hand drawings, and, as we believe, can be very useful to increase visibility of the work. This has proven to offer a nice opportunity for exposure i.p. for the first author(s) of the study. Please see the following link for representative examples and their integration into the article web page:

<https://www.embopress.org/doi/full/10.15252/emj.2019103932>

If you have any questions, please do not hesitate to call or email the Editorial Office.

Best regards,

Daniel Klimmeck

Daniel Klimmeck, PhD
Senior Editor
The EMBO Journal
EMBO
Postfach 1022-40
Meyerhofstrasse 1
D-69117 Heidelberg
contact@embojournal.org
Submit at: <http://emboj.msubmit.net>